# Optimal Transport-Induced Samples against Out-of-Distribution Overconfidence

**Keke Tang**[1*], **Ziyong Du**[1*], **Xiaofei Wang**[2,3†], **Weilong Peng**[1], **Peican Zhu**[4†], **Zhihong Tian**[1,5]
[1]Guangzhou University, [2]SmartMore Corporation
[3]University of Science and Technology of China, [4]Northwestern Polytechnical University
[5]Guangdong Key Laboratory of Industrial Control System Security
`tangbohutbh@gmail.com, duxiaoshuaicst@gmail.com`
`wxf9545@mail.ustc.edu.cn, wlpeng@tju.edu.cn`
`ericcan@nwpu.edu.cn, tianzhihong@gzhu.edu.cn`

## Abstract

Deep neural networks (DNNs) often produce overconfident predictions on out-of-distribution (OOD) inputs, undermining their reliability in open-world environments. Singularities in semi-discrete optimal transport (OT) mark regions of semantic ambiguity, where classifiers are particularly prone to unwarranted high-confidence predictions. Motivated by this observation, we propose a principled framework to mitigate OOD overconfidence by leveraging the geometry of OT-induced singular boundaries. Specifically, we formulate an OT problem between a continuous base distribution and the latent embeddings of training data, and identify the resulting singular boundaries. By sampling near these boundaries, we construct a class of OOD inputs, termed optimal transport-induced OOD samples (OTIS), which are geometrically grounded and inherently semantically ambiguous. During training, a confidence suppression loss is applied to OTIS to guide the model toward more calibrated predictions in structurally uncertain regions. Extensive experiments show that our method significantly alleviates OOD overconfidence and outperforms state-of-the-art methods.

## 1 Introduction

Deep neural networks (DNNs) have achieved remarkable success in classification problems (He et al., 2015; Tang et al., 2025b). However, they are typically trained and evaluated under the closed-world assumption that all test inputs are drawn from the same distribution as the training data (Bendale & Boult, 2015). In open-world scenarios, this assumption often breaks down, as inputs from previously unseen classes or domains naturally arise. When confronted with such out-of-distribution (OOD) inputs, DNNs tend to produce confident predictions despite lacking relevant experience. This overconfidence has been extensively documented (Nguyen et al., 2015; Goodfellow et al., 2015), and poses serious risks in safety-critical applications. Mitigating such overconfidence is therefore a fundamental step toward reliable deployment of DNN systems in open-world environments.

To address OOD overconfidence, a widely adopted strategy is to perform OOD detection at test time, where confidence-based scoring functions are used to separate in-distribution (ID) and OOD inputs (Hendrycks & Gimpel, 2017; Liang et al., 2018; Tang et al., 2023a;b; 2024; 2025a; Zhao et al., 2025; Yang et al., 2025; Fang et al., 2024; 2025a;b;c). However, such post-hoc filtering merely avoids high-confidence mistakes without fundamentally altering the model's tendency to produce overconfident predictions on unfamiliar inputs. As a more proactive alternative, recent work has explored exposing the model to proxy OOD samples during training, encouraging it to produce low-confidence predictions on inputs outside the training distribution. These proxy samples are typically constructed using external datasets (Hendrycks et al., 2018), input corruption (Hein et al., 2019), class mixing (Tang et al., 2021), or latent outlier synthesis (Du et al., 2022). Despite their empirical success, these heuristically designed samples often lack theoretical grounding and fail to target semantically ambiguous regions where overconfidence is most likely to occur.

---

*Equal contribution †Corresponding authors

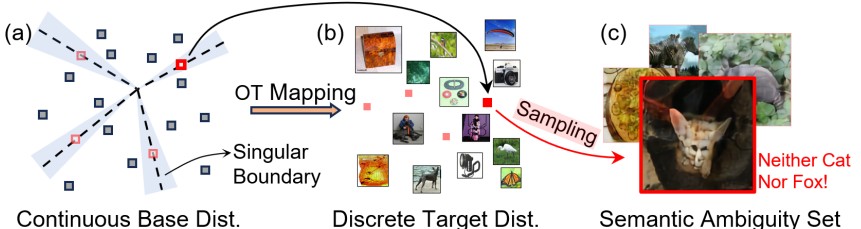

Figure 1: Given a semi-discrete optimal transport (OT) map from (a) a continuous base distribution to (b) a discrete target distribution over images, the singular boundaries in the source domain are mapped to (c) the semantic ambiguity set in image space, which typically contains images with features from multiple classes.

A promising direction to overcome these limitations is to exploit the geometric insights offered by optimal transport (OT) (Villani et al., 2008; Santambrogio, 2015; Mérigot, 2011), which provides a principled framework for understanding data structure and distributional uncertainty. In the semi-discrete OT setting, a continuous source distribution (e.g., Gaussian noise) is mapped onto a discrete target measure via the gradient of a convex potential. This potential induces a partition of the source domain into convex regions, each assigned to a discrete target point. The interfaces between adjacent regions correspond to non-differentiable loci of the potential, known as transport singularities, where the direction of transport changes abruptly (Figalli, 2010; An et al., 2020). These singular boundaries mark structurally unstable zones in the transport map and often coincide with semantically ambiguous regions in classification models, where decision behavior becomes less reliable (see Fig. 1). As such, these regions provide a theoretically grounded signal for identifying where overconfidence may arise, offering a compelling basis for model regularization.

Building on this insight, we propose a principled framework for mitigating OOD overconfidence by exploiting the geometric singularities arising in semi-discrete optimal transport. Our approach first encodes ID samples into a compact latent space via an autoencoder. A semi-discrete OT problem is then solved between a continuous base distribution, e.g., a Gaussian or uniform distribution, and the discrete set of latent embeddings, inducing a convex partition of the latent space. Among the adjacent regions, those with large angular deviations in transport direction are identified as structurally unstable zones. To construct ambiguous inputs, we interpolate between the centroids of neighboring regions associated with these zones, and decode the resulting latent vectors back to the input space. The generated samples, termed optimal transport-induced OOD samples (OTIS), are used during training with a confidence suppression loss, guiding the model to produce low-confidence predictions on structurally ambiguous regions. Extensive experiments across multiple architectures and diverse ID/OOD settings demonstrate that our method effectively reduces OOD overconfidence without sacrificing ID accuracy, consistently outperforming state-of-the-art approaches.

Overall, our contribution is summarized as follows:

- We establish a theoretical link between geometric singularities in semi-discrete optimal transport and the emergence of overconfident predictions on OOD inputs.
- We develop a novel OOD overconfidence mitigation framework that regularizes model confidence using semantically ambiguous samples derived from OT-induced singularities.
- We demonstrate through experiments across multiple architectures and ID/OOD settings that our approach outperforms state-of-the-art methods in mitigating OOD overconfidence.

## 2 PROBLEM FORMULATION AND MOTIVATION

### 2.1 OOD OVERCONFIDENCE

**Preliminaries on OOD Overconfidence Issue.** We consider a standard multi-class classification task with label space $\{1, 2, \ldots, K\}$. Let $\mathcal{A}$ denote the space of all candidate inputs, and let $\mathcal{I} \subseteq \mathcal{A}$ be the in-distribution (ID) subset. The set of out-of-distribution (OOD) inputs is given by $\mathcal{A} \setminus \mathcal{I}$. A classifier $f : \mathcal{A} \to \{1, \ldots, K\}$ is trained on $\mathcal{I}$, but may produce overconfident predictions on inputs from $\mathcal{A} \setminus \mathcal{I}$, compromising reliability in open-world deployments.

**Mitigating Overconfidence via Suppressing Predictions on Proxy OOD Samples.** A widely adopted strategy to mitigate OOD overconfidence is to expose the model to a set of proxy OOD samples $\mathcal{O} \subseteq \mathcal{A} \setminus \mathcal{I}$ during training. These samples are typically constructed by injecting noise, applying input corruptions (Hein et al., 2019), performing class mixing (Tang et al., 2021), or sampling from unrelated datasets (Hendrycks et al., 2018). The objective is to encourage the model to produce low-confidence predictions on unfamiliar inputs. The corresponding regularization loss is often formulated as

$$\mathcal{L}_{\text{proxy}} = \mathbb{E}_{x \in \mathcal{O}}[s(f(x))], \tag{1}$$

where $s(f(x))$ denotes a confidence score (e.g., the maximum softmax probability). However, existing methods typically construct $\mathcal{O}$ with heuristic rules and don't explicitly target structurally unstable or semantically ambiguous regions where overconfident predictions are most likely to occur.

## 2.2 SEMI-DISCRETE OPTIMAL TRANSPORT

Semi-discrete optimal transport (OT) models the alignment between a continuous source measure and a discrete target distribution. It naturally induces a geometric partition of the source domain, providing a structured view of how samples are assigned to semantic prototypes.

**Preliminaries on Semi-Discrete Optimal Transport.** Let $\mu$ be an absolutely continuous probability measure supported on a convex domain $\Omega \subseteq \mathbb{R}^d$, such as a Gaussian or uniform distribution. Let the target domain be a discrete set $\mathcal{Y} = \{y_1, \ldots, y_n\} \subseteq \mathbb{R}^d$ equipped with a probability measure $\nu = \sum_{i=1}^n w_i \delta_{y_i}$, where $w_i \geq 0$ and $\sum_i w_i = 1$.

The semi-discrete optimal transport problem seeks a measurable map $T : \Omega \to \mathbb{R}^d$ that pushes $\mu$ onto $\nu$, i.e., $T_{\#}\mu = \nu$, and minimizes the expected transport cost:

$$\int_\Omega \frac{1}{2} \|T(z) - z\|^2 \, d\mu(z), \tag{2}$$

where $z \in \Omega$ denotes a source point.

According to Brenier's theorem (Brenier, 1991), if $\mu$ is absolutely continuous, the optimal transport map $T$ exists and is given by the gradient of a convex function:

$$T(z) = \nabla u_{\mathbf{h}}(z), \quad u_{\mathbf{h}}(z) = \max_i \{\langle y_i, z \rangle + h_i\}, \tag{3}$$

where $\mathbf{h} = \{h_i\}$ are scalar offsets. The function $u_{\mathbf{h}}$ is the upper envelope of affine functions, forming a piecewise-linear convex surface over $\Omega$. To ensure identifiability, a normalization condition such as $\sum_i h_i = 0$ is typically imposed.

**Partition Boundaries Induced by Optimal Transport.** The convex structure of $u_{\mathbf{h}}$ induces a partition of the continuous source domain $\Omega$ into convex Laguerre cells, such that $\Omega = \bigcup_i \mathcal{W}_i$, where

$$\mathcal{W}_i = \{z \in \Omega \mid \langle y_i, z \rangle + h_i \geq \langle y_j, z \rangle + h_j, \ \forall j \neq i\}. \tag{4}$$

These cells form a power diagram over $\Omega$, generalizing Voronoi diagrams by incorporating the offsets $\{h_i\}$. Each adjacent pair $(\mathcal{W}_i, \mathcal{W}_j)$ shares a boundary hyperplane:

$$\mathcal{S}_{ij} = \{z \in \Omega \mid \langle a_{ij}, z \rangle + b_{ij} = 0\}, \quad a_{ij} = y_i - y_j, \quad b_{ij} = h_i - h_j. \tag{5}$$

These transport-induced boundaries delineate semantic transitions and provide the foundation for constructing structurally ambiguous samples.

## 2.3 OT SINGULARITIES AS SOURCES OF SEMANTICALLY AMBIGUOUS OOD SAMPLES

According to the regularity theory of semi-discrete optimal transport (Figalli, 2010; Chen & Figalli, 2017), the Brenier potential $u_{\mathbf{h}}$ becomes non-differentiable on a singular set $\Sigma \subset \Omega$ when the target measure $\nu$ is multimodal or disconnected. The transport map $T(z) = \nabla u_{\mathbf{h}}(z)$ is discontinuous across $\Sigma$, leading to abrupt changes in transport direction.

Although $\Sigma$ has measure zero, its neighborhood contains inputs near multiple transport boundaries, where assignments are unstable and semantic alignment is ambiguous. These regions often correspond to high-confidence mispredictions and indicate heightened robustness risk.

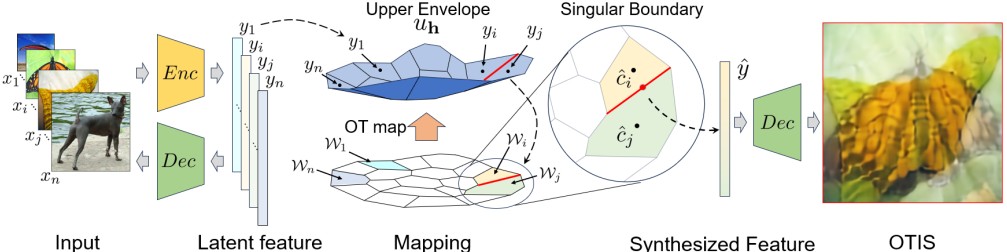

Figure 2: Overview of our framework for generating OTIS. Input images are encoded into a latent space, where a semi-discrete optimal transport (OT) map establishes a power-diagram partition with its convex potential visualized as the upper envelope. Singularity boundaries identified from this OT map guide the generation of interpolated latent features, which are decoded to synthesize OTIS.

Therefore, by setting the OT target distribution to the support of training images, we can construct transport-induced singularities and map from their vicinity back to the input space, yielding semantically ambiguous samples near class transitions. These samples serve as proxies for OOD uncertainty and offer a principled basis for training-time regularization to mitigate OOD overconfidence.

## 3 METHOD

This section presents our framework for alleviating OOD overconfidence by exploiting the structural properties of OT-induced boundaries introduced in Sec. 2.2. The proposed pipeline first encodes input samples into a compact latent space, then constructs optimal transport-induced OOD samples (OTIS) based on the geometry of the OT partition, and finally mitigates overconfident predictions by training the classifier with these ambiguity-aware samples. Please refer to Fig. 2 for demonstration.

### 3.1 LATENT REPRESENTATION FOR OT

To facilitate optimal transport modeling in a compact geometry, we construct a latent representation space $\mathcal{Y} \subseteq \mathbb{R}^d$ using an autoencoder. Each ID sample $x \in \mathcal{I}$ is encoded by a neural encoder $Enc$ into a latent vector:

$$y = Enc(x). \tag{6}$$

The resulting $y$ lies in the latent space $\mathcal{Y}$, which is designed to be geometrically structured and compact. To enable reconstruction and downstream training, a decoder $Dec$ maps latent vectors back to the input domain:

$$x^{'} = Dec(y). \tag{7}$$

We denote by $\{x_i\}_{i=1}^n \subset \mathcal{I}$ a set of $n$ ID training samples, and let their latent embeddings $\{y_i = Enc(x_i)\}_{i=1}^n$ serve as support points for the discrete target measure $\nu = \sum_i w_i \delta_{y_i}$ in the semi-discrete optimal transport formulation. Modeling in latent space improves regularity and tractability compared to operating in the original input domain.

### 3.2 CONSTRUCTION OF OTIS

**Partition Estimation via Potential Optimization.** Building on the semi-discrete OT formulation introduced in Sec. 2.2, we aim to construct the transport-induced partition in the latent space by estimating the $\mu$-volume of each Laguerre cell. These volumes characterize how the source measure distributes mass across regions associated with the support points $\{y_i\}$.

We estimate the $\mu$-volume of each Laguerre cell $\mathcal{W}_i(\mathbf{h})$ using a Monte Carlo strategy. Specifically, we sample $M$ latent points $\{z_j\}_{j=1}^M \sim \mu$ and assign each point to the region that maximizes the potential: $i^*(z_j) = \arg\max_k \{\langle y_k, z_j \rangle + h_k\}$. The proportion of points assigned to each region yields an empirical estimate of its $\mu$-volume: $\hat{\mu}(\mathcal{W}_i(\mathbf{h})) = \frac{\#\{j | i^*(z_j) = i\}}{M}$.

To compute the optimal offsets $\mathbf{h}$, we minimize the discrepancy between estimated $\mu$-volumes and a target measure $\nu = \sum_i w_i \delta_{y_i}$, using uniform weights $w_i = 1/n$:

$$\min_{\mathbf{h} \in \mathbb{R}^n} \ \mathcal{E}(\mathbf{h}) = \sum_{i=1}^{n} \left( \hat{\mu}(\mathcal{W}_i(\mathbf{h})) - w_i \right)^2, \quad \text{s.t.} \ \sum_i h_i = 0. \tag{8}$$

After convergence, each region pair $(y_i, y_j)$ defines a boundary $\mathcal{S}_{ij}$, and we collect $\mathcal{S} = \bigcup_{i<j} \mathcal{S}_{ij}$ as candidate boundary set.

**Identification of Singular Boundaries.** We operationalize the concept of transport singularities by assigning a geometric score to each OT-induced boundary and selecting those most likely to exhibit discontinuous or unstable behavior. Given the candidate boundary set $\mathcal{S}$ constructed from adjacent support points in the OT partition, we quantify the angular deviation across each boundary $\mathcal{S}_{ij}$ as

$$\text{score}(\mathcal{S}_{ij}) = \arccos \left( \frac{\langle y_i, \ y_j \rangle}{\|y_i\| \cdot \|y_j\|} \right). \tag{9}$$

This score reflects the change in transport direction between adjacent cells. Larger values suggest sharper directional shifts and greater likelihood of singularity. We sort all boundaries by score and retain a fixed top-ranked proportion to form the singular boundary set $\mathcal{S}' \subseteq \mathcal{S}$, which is used to guide ambiguity-aware sample generation in the latent space.

**Synthesis of Optimal Transport-induced OOD Samples (OTIS).** For each selected boundary $\mathcal{S}_{ij} \in \mathcal{S}'$, we estimate the mass centers $\hat{c}_i$ and $\hat{c}_j$ of the corresponding Laguerre cells $\mathcal{W}_i$ and $\mathcal{W}_j$ via Monte Carlo approximation. Specifically, each center is computed as:

$$\hat{c}_t = \frac{1}{\#\{z_k \in \mathcal{W}_t\}} \sum_{z_k \in \mathcal{W}_t} z_k, \quad \text{with } z_k \sim \mu, \ t \in \{i, j\}. \tag{10}$$

We then sample a latent point $z \sim \mu$ and compute inverse-distance interpolation weights:

$$\lambda_i = \frac{1/\|z - \hat{c}_i\|}{1/\|z - \hat{c}_i\| + 1/\|z - \hat{c}_j\|}, \quad \lambda_j = 1 - \lambda_i. \tag{11}$$

To ensure continuity near transport-induced boundaries, we adopt a smoothed transport extension $\tilde{T}(\cdot)$, defined as:

$$\hat{y} = \tilde{T}(z) = \lambda_i T(\hat{c}_i) + \lambda_j T(\hat{c}_j). \tag{12}$$

This extension provides a softened approximation of the discrete OT map in structurally ambiguous regions, reducing aliasing artifacts and enhancing the semantic coherence of generated features.

Finally, we decode $\hat{y}$ via $\hat{x} = Dec(\hat{y})$, and include it in the proxy OOD set $\mathcal{O}_{\text{OT}}$. These samples, termed optimal transport-induced OOD samples (OTIS), are used during training to suppress over-confident predictions near OT-induced singularities.

### 3.3 Training with OTIS for Mitigating OOD Overconfidence

To mitigate OOD overconfidence, we incorporate OTIS, $\hat{x} \in \mathcal{O}_{\text{OT}}$, which serves as proxy OOD inputs generated near OT-induced singular boundaries, into the training of DNNs. We regularize model predictions on these inputs using a confidence suppression loss following (Tang et al., 2021):

$$\mathcal{L}_{\text{sup}}(\hat{x}) = \sum_{i=1}^{K} \frac{1}{K} \log V_i(\hat{x}), \tag{13}$$

where $V_i(\hat{x})$ denotes the softmax probability for category $i$. This loss encourages the model to distribute its confidence evenly across all classes, thereby enforcing averaged uncertainty and suppressing overconfident predictions on ambiguous inputs.

During training, each batch consists of 50% ID samples supervised with cross-entropy loss and 50% OTIS guided by the suppression loss in Eqn. 13.

Table 1: Comparison of eight methods in mitigating OOD overconfidence. We report test error (TE), mean maximum confidence on ID (ID MMC ↑) and OOD (OOD MMC ↓) inputs. All values are in percent (%). Note: OE and CCUd leverage auxiliary datasets, while others do not.

| ID | metric | OOD | - | CEDA | ACET | CCUs | CODES | VOS | Ours | OE | CCUd |
|---|---|---|---|---|---|---|---|---|---|---|---|
| | with auxiliary dataset | | | | | | | | | ✓ | ✓ |
| CIFAR-10 | TE | CIFAR-10 | 5.79 | 8.55 | 6.87 | 5.86 | 7.73 | 5.81 | 7.52 | 6.80 | **5.55** |
| | ID MMC | CIFAR-10 | **97.98** | 96.06 | 96.84 | 97.50 | 93.46 | 97.14 | 95.46 | 96.80 | 97.30 |
| | OOD MMC | SVHN | 84.22 | 71.62 | 82.16 | 72.48 | 72.35 | 73.16 | **13.18** | 55.82 | 76.52 |
| | | CIFAR-100 | 85.98 | 80.18 | 82.36 | 75.95 | 74.69 | 81.04 | **64.79** | 80.94 | 81.49 |
| | | LSUN_CR | 77.18 | 60.70 | 65.15 | 55.93 | 51.64 | 70.5 | **30.36** | 68.61 | 79.49 |
| | | Textures_C | 86.28 | 72.47 | 78.58 | 59.01 | 61.32 | 78.5 | **48.75** | 67.66 | 74.35 |
| | | Noise | 85.55 | 60.51 | 64.74 | 50.61 | 43.70 | 51.38 | 16.18 | **10.13** | 74.95 |
| | | Uniform | 87.45 | 10.04 | **10.00** | **10.00** | 11.13 | 80.65 | **10.00** | 26.20 | **10.00** |
| | | Adv. Noise | 98.90 | 43.04 | **10.00** | **10.00** | 37.66 | 95.56 | 26.42 | 91.15 | **10.00** |
| | | Adv. Samples | 95.45 | 69.44 | **26.27** | 84.20 | 92.37 | 95.85 | 57.71 | 88.86 | 98.00 |
| CIFAR-100 | TE | CIFAR-100 | **23.03** | 30.96 | 25.95 | 50.66 | 25.68 | 23.56 | 27.73 | 25.63 | 23.83 |
| | ID MMC | CIFAR-100 | 82.96 | 80.36 | 83.75 | 56.43 | **87.76** | 85.41 | 83.90 | 84.02 | 82.01 |
| | OOD MMC | SVHN | 48.27 | 63.03 | 62.85 | 65.49 | 66.11 | 58.76 | **9.30** | 52.57 | 58.62 |
| | | CIFAR-10 | 56.34 | 62.65 | 65.36 | **45.70** | 68.96 | 62.29 | 61.88 | 65.01 | 56.34 |
| | | LSUN_CR | 55.01 | 51.34 | 50.35 | 51.82 | 41.85 | 58.02 | **40.64** | 50.11 | 56.98 |
| | | Textures_C | 57.42 | 66.69 | 66.26 | 58.30 | 63.77 | 61.96 | **55.89** | 60.66 | 56.14 |
| | | Noise | 92.17 | 82.99 | 78.94 | 68.00 | 80.15 | 92.92 | 71.61 | **1.07** | 70.59 |
| | | Uniform | 74.76 | 1.09 | **1.00** | **1.00** | 1.10 | 72.44 | **1.00** | 3.25 | 7.38 |
| | | Adv. Noise | 86.47 | 23.58 | **1.00** | **1.00** | 27.85 | 77.58 | 8.94 | 67.36 | 46.52 |
| | | Adv. Samples | 92.86 | 19.35 | **2.25** | 51.16 | 87.88 | 84.01 | 13.71 | 47.75 | 63.88 |
| SVHN | TE | SVHN | 3.68 | 4.66 | 3.50 | 8.42 | 4.29 | 3.51 | 4.83 | 3.53 | **3.17** |
| | ID MMC | SVHN | 98.41 | 97.17 | 97.78 | 90.64 | 93.35 | 97.99 | 96.25 | 97.67 | **98.50** |
| | OOD MMC | CIFAR-10 | 75.15 | 73.70 | 62.54 | **46.92** | 61.09 | 71.39 | 61.37 | 62.48 | 66.36 |
| | | CIFAR-100 | 75.36 | 74.01 | 63.42 | **44.81** | 54.09 | 72.5 | 56.88 | 62.44 | 66.12 |
| | | LSUN_CR | 66.75 | 66.44 | 50.98 | 49.99 | 56.11 | 64.39 | **48.85** | 54.52 | 75.41 |
| | | Textures_C | 74.22 | 73.03 | 50.30 | 51.55 | **22.77** | 71.58 | 51.20 | 45.76 | 56.06 |
| | | Noise | 98.41 | 97.17 | 97.78 | 97.64 | 93.35 | 97.99 | **92.42** | 97.67 | 98.00 |
| | | Uniform | 75.43 | 10.21 | **10.00** | **10.00** | 10.25 | 71.32 | **10.00** | 10.08 | **10.00** |
| | | Adv. Noise | 93.75 | 80.21 | **10.00** | **10.00** | 25.59 | 91.99 | 24.53 | 21.88 | **10.00** |
| | | Adv. Samples | 89.05 | 84.83 | 71.20 | 82.33 | 68.05 | 88.13 | 66.81 | **42.23** | 99.96 |
| MNIST | TE | MNIST | 0.67 | 0.82 | 0.72 | 1.28 | 0.66 | 0.68 | 1.00 | 0.83 | **0.61** |
| | ID MMC | MNIST | 99.47 | 99.32 | 99.27 | 98.41 | **99.65** | 99.43 | 98.98 | 99.30 | 99.18 |
| | OOD MMC | FMNIST | 58.80 | 55.62 | 40.76 | 32.09 | 57.29 | 62.21 | **20.05** | 40.83 | 34.88 |
| | | EMNIST | 82.18 | 82.40 | 79.12 | 78.34 | 86.02 | 81.85 | **50.01** | 79.87 | 80.66 |
| | | GrayCIFAR | 41.41 | 29.35 | 12.80 | **10.01** | 25.98 | 39.85 | 15.48 | 13.76 | 10.02 |
| | | Kylberg | 26.56 | 10.42 | **10.00** | **10.00** | 10.09 | 19.48 | **10.00** | 10.17 | **10.00** |
| | | Noise | 10.95 | 10.62 | 10.01 | 10.30 | 10.22 | 11.25 | **10.00** | 10.07 | 10.91 |
| | | Uniform | 27.20 | 10.25 | **10.00** | **10.00** | 10.01 | 19.56 | **10.00** | 10.13 | **10.00** |
| | | Adv. Noise | 95.08 | 75.67 | **10.00** | **10.00** | 65.47 | 19.33 | **10.00** | 25.50 | **10.00** |
| | | Adv. Samples | 89.09 | 88.01 | 59.03 | 10.27 | 93.77 | 87.75 | **10.00** | 76.37 | **10.00** |
| FMNIST | TE | FMNIST | 6.91 | 6.10 | 7.52 | 11.51 | 7.25 | 5.98 | 7.38 | 6.46 | **4.84** |
| | ID MMC | FMNIST | 97.01 | 97.48 | 94.49 | 88.53 | 97.40 | 97.31 | 95.64 | 98.02 | **98.32** |
| | OOD MMC | MNIST | 77.89 | 75.83 | 80.80 | 48.84 | 89.39 | 76.46 | **44.63** | 75.86 | 70.20 |
| | | EMNIST | 81.32 | 77.58 | 85.56 | **33.64** | 90.43 | 77.28 | 47.93 | 78.64 | 67.16 |
| | | GrayCIFAR | 90.73 | 56.79 | 32.01 | 11.37 | 38.54 | 82.86 | **11.14** | 16.63 | 45.66 |
| | | Kylberg | 94.25 | 17.30 | 15.25 | 10.93 | 17.11 | 81.92 | **10.35** | 18.43 | 59.11 |
| | | Noise | 92.37 | 35.67 | 11.27 | 12.76 | 13.32 | 64.16 | **10.00** | 10.02 | 52.77 |
| | | Uniform | 85.78 | 10.03 | **10.00** | **10.00** | 10.31 | 66.53 | **10.00** | 32.88 | **10.00** |
| | | Adv. Noise | 99.76 | 16.98 | **10.00** | **10.00** | 19.64 | 98.69 | 19.39 | 97.57 | **10.00** |
| | | Adv. Samples | 89.64 | 67.10 | 50.20 | 98.60 | 51.22 | 83.71 | **15.09** | 40.81 | 99.98 |

# 4 EXPERIMENTS

## 4.1 EXPERIMENTAL SETUP

**Implementation.** For latent representation, we train an autoencoder using a five-layer convolutional encoder with 512 channels paired with a symmetric five-layer transposed convolutional decoder for small images ($28 \times 28$ or $32 \times 32$), resulting in a latent dimensionality of 256. For ImageNet ($224 \times 224$), a symmetric VGG-16 architecture is adopted as the autoencoder, yielding a latent dimensionality of 1024. All models are trained for 200 epochs using the Adam optimizer with a learning rate of 0.0001. The semi-discrete OT problem is solved following (An et al., 2020).

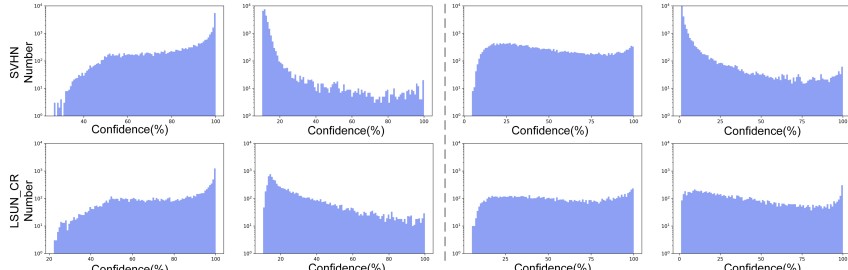

Figure 3: Histograms of maximum confidence scores on OOD inputs before and after applying our method. Results are shown for ResNet-18 trained on CIFAR-10 (left) and CIFAR-100 (right).

We select the top 10% of boundaries with the largest singular scores for OTIS generation and set the weight $\lambda_i$ according to Eqn. 11. The overall training procedure for confidence suppression follows the default configuration of CCU (Meinke & Hein, 2020). All experiments are implemented in PyTorch (Paszke et al., 2019) and conducted on a workstation with eight NVIDIA RTX 4090 GPUs.

**ID/OOD Configurations.** For low-resolution benchmarks, we use CIFAR-10, CIFAR-100 (Krizhevsky et al., 2009), SVHN (Netzer et al., 2011), MNIST, and FashionMNIST (FM-NIST) (Xiao et al., 2017) as ID datasets. Each can also serve as an OOD dataset when excluded from training, enabling diverse ID/OOD configurations. Additional OOD datasets include LSUN_CR (a classroom subset of LSUN (Yu et al., 2015)), Textures_C (cropped from Textures (Ashworth & McLellan, 1985)), EMNIST (Cohen et al., 2017), GrayCIFAR (grayscale CIFAR-10), and Kylberg (Kylberg, 2011). We also include Noise and Uniform samples following the procedure in (Meinke & Hein, 2020). Furthermore, we evaluate on Adversarial Noise (obtained by maximizing classifier confidence near random inputs) and Adversarial Samples (constructed near ID inputs but off the data manifold) (Meinke & Hein, 2020). For high-resolution evaluation, we use ImageNet (Deng et al., 2009) as the ID dataset, and consider OpenImage-O (Kuznetsova et al., 2020), iNaturalist (Van Horn et al., 2018), SUN (Xiao et al., 2010), Places (Zhou et al., 2016), and Textures (Ashworth & McLellan, 1985) as OOD benchmarks.

**Baselines.** On low-resolution datasets, we compare our method with five approaches that do not rely on auxiliary OOD data: CEDA and ACET (Hein et al., 2019), CCUs (Meinke & Hein, 2020) (a noise-based variant of CCU), CODES (Tang et al., 2021), and VOS (Du et al., 2022). We also include two methods that utilize external datasets: OE (Hendrycks et al., 2018), and CCUd, a variant of CCU trained with the 80 Million Tiny Images dataset (Torralba et al., 2008), excluding all samples overlapping with CIFAR-10/100, following (Meinke & Hein, 2020). For high-resolution experiments on ImageNet, we compare against CEDA, ACET, CODES, and VOS as baselines.

**Setup and Evaluation Metrics.** We use LeNet for MNIST and ResNet-18 for CIFAR-10, CIFAR-100, SVHN, and FMNIST. For high-resolution datasets, ResNet-50 is employed. All models are evaluated on their respective ID test sets to report test error (TE), and on both ID and OOD datasets to compute mean maximum confidence (MMC), following the protocol of (Meinke & Hein, 2020). To further assess the model's ability to suppress overconfident predictions on unfamiliar inputs, we also report the false positive rate at 95% true positive rate (FPR95), where confidence serves as the scoring function to distinguish between ID and OOD samples.

## 4.2 MAIN RESULTS

**Results on Low-resolution Benchmarks.** Tab. 1 reports the performance of our method and prior approaches across five datasets. Our method consistently outperforms all baselines in mitigating OOD overconfidence, achieving the lowest OOD MMC in nearly all settings while maintaining strong ID performance. On CIFAR-10 and CIFAR-100, we obtain significantly lower OOD MMCs (e.g., 13.18% on SVHN and 9.30% on CIFAR-100) than all competitors, including OE and CCUd which rely on large auxiliary datasets. Meanwhile, test errors and ID MMC remain competitive, indicating no sacrifice in ID accuracy. On MNIST and FMNIST, our approach achieves near-optimal suppression (e.g., 10.00% on Noise and Uniform) and outperforms prior methods even on challenging shifts such as GrayCIFAR and EMNIST. For adversarial shifts, our method again delivers superior performance (e.g., 8.94% on CIFAR-100 and 15.09% on FMNIST), without relying on any

Table 2: Comparison of five methods in mitigating OOD overconfidence for ResNet-50 on ImageNet. We report test error (TE), mean maximum confidence (MMC; ID ↑, OOD ↓), and FPR95 ↓.

| Method | ID | | OOD | | | | | | | | | |
| | ImageNet | | OpenImage-O | | iNaturalist | | SUN | | Places | | Textures | |
| | TE | MMC | MMC | FPR95 | MMC | FPR95 | MMC | FPR95 | MMC | FPR95 | MMC | FPR95 |
|---|---|---|---|---|---|---|---|---|---|---|---|---|
| - | **23.87** | 79.70 | 43.84 | 66.84 | 35.84 | 52.77 | 46.15 | 68.58 | 48.33 | 71.57 | 46.51 | 66.13 |
| CEDA | 24.72 | 79.87 | 44.37 | 67.50 | 34.76 | 50.67 | 46.25 | 68.36 | 47.77 | 70.83 | 47.18 | 66.33 |
| ACET | 25.15 | 78.91 | 43.04 | 67.76 | 35.31 | 54.39 | 46.18 | 71.02 | 47.17 | 72.40 | 46.49 | 68.33 |
| CODES | 27.71 | 81.39 | **34.25** | 65.26 | 32.87 | 56.38 | 39.36 | 70.23 | 46.70 | 68.10 | 45.80 | **62.62** |
| VOS | 24.73 | 82.60 | 37.84 | 67.67 | 39.04 | 52.12 | 39.26 | **66.42** | 51.02 | 70.72 | 47.62 | 63.58 |
| Ours | 26.43 | **88.93** | 34.96 | **63.96** | **31.01** | **49.16** | **38.80** | 68.17 | **45.78** | 68.08 | **38.81** | 66.50 |

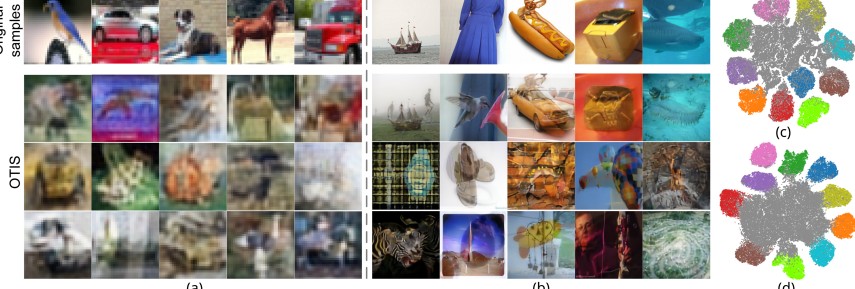

Figure 4: Visualizations of original inputs and corresponding OTIS from (a) CIFAR-10 and (b) ImageNet. t-SNE plots of inputs and corresponding OTIS from (c) FMNIST and (d) CIFAR-10.

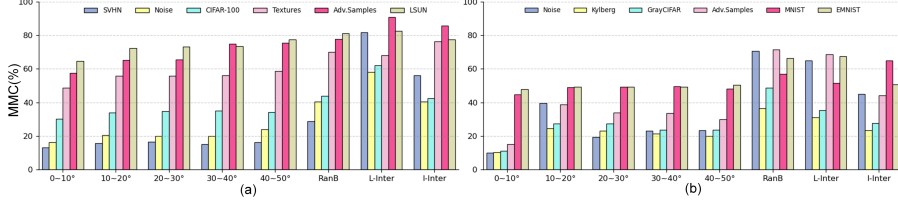

Figure 5: OOD confidence of ResNet-18 trained on (a) CIFAR-10 and (b) FMNIST under different sampling strategies. Boundary-based methods use top-$k$% singularities or random boundaries (RanB). L-Inter and I-Inter denote latent and image-level interpolation without boundary guidance.

handcrafted augmentations or external supervision. Overall, our method consistently achieves the best overall trade-off between accuracy and confidence calibration across diverse OOD conditions.

**Visualization of Confidence Distributions.** Fig. 3 shows histograms of maximum softmax confidence scores on OOD inputs (SVHN and LSUN_CR) for ResNet-18 models trained on CIFAR-10 and CIFAR-100. Compared to the baseline, our method substantially reduces the prevalence of high-confidence predictions, shifting the distributions toward lower confidence values. This illustrates the effectiveness of our approach in alleviating OOD overconfidence.

**Qualitative Analysis of OTIS.** As shown in Fig. 4, OTIS instances exhibit semantically ambiguous yet visually coherent content, often obscuring discriminative features and reflecting structural uncertainty aligned with our objective. The t-SNE (Maaten & Hinton, 2008) visualizations further reveal that OTIS instances cluster in inter-class transition zones rather than collapsing onto any single class manifold. This validates their role as structurally grounded inputs for suppressing overconfident predictions near decision boundaries.

**ImageNet Results.** Tab. 2 presents results on ImageNet with ResNet-50. Our method achieves the highest ID MMC (88.93%) and significantly reduces OOD confidence, attaining the lowest OOD MMC on most benchmarks. It also achieves the best FPR95 on key datasets such as iNaturalist (49.16%) and OpenImage-O (63.96%). Although the test error (26.43%) is slightly higher than some baselines, the overall performance demonstrates a favorable trade-off between ID accuracy and OOD calibration, outperforming CEDA, ACET, CODES, and VOS in most cases.

### 4.3 ABLATION STUDIES AND OTHER ANALYSIS

**Effectiveness of High-Singularity Boundaries.** We investigate whether sampling near boundaries with higher singularity scores improves OOD regularization. As shown in Fig. 5, selecting the top

Figure 6: ROC curves for OOD detection on four ID→OOD settings using MSP, ODIN, and ReAct, with and without applying our training framework.

10% of boundaries consistently achieves the lowest OOD MMC across multiple datasets, indicating that these regions are most effective in capturing structural ambiguity and eliciting overconfidence. Performance gradually degrades when selecting lower-ranked boundaries, but remains superior to random boundary selection (RanB), confirming the benefit of singularity-aware sampling.

**Singularity-Based Sampling vs. Interpolation Baselines.** We compare boundary-based sampling with interpolation strategies that do not leverage OT geometry. As shown in Fig. 5, both latent-space (L-Inter) and input-space (I-Inter) interpolation yield substantially higher OOD MMC than boundary-based methods. This suggests that naïve mixing fails to produce semantically ambiguous inputs effective for regularizing overconfidence. In contrast, OT-induced singular boundaries offer geometrically grounded, uncertain regions that serve as more informative training signals.

**Performance in Improving OOD Detection.** To assess whether our confidence regularization enhances standard OOD detection, we apply MSP (Hendrycks & Gimpel, 2017), ODIN (Liang et al., 2018), and ReAct (Sun et al., 2021), with and without training the classifier using our OTIS-based suppression loss. As shown in Fig. 6, models trained with our framework (MSP+Ours, ODIN+Ours, ReAct+Ours) consistently exhibit ROC curves that dominate their respective baselines, indicating improved true positive rates across a range of false positive thresholds in all test cases. The improvement is especially pronounced under challenging shifts such as CIFAR-10→Textures_C. These results confirm that suppressing overconfidence in structurally ambiguous regions during training yields more calibrated models and enables more effective OOD detection at inference.

**Why OTIS Instead of Other Heuristic Boundary Samples?** Tab. 3 reports Fréchet inception distance (FID) to CIFAR-10 and the mean maximum confidence (MMC) of a ResNet-18 classifier trained on CIFAR-10, evaluated on the CIFAR-10 test set and on the boundary samples used by CEDA, ACET, CODES, VOS, OE, and OTIS. Among these sample sets, OTIS achieves the smallest FID while still inducing high MMC, meaning that OTIS samples stay closest to the CIFAR-10 ID manifold and are predicted with high confidence. Although ACET attains the highest MMC, the Adv. Noise it uses lies much farther from the ID manifold, as reflected by its much larger FID. Overall, OTIS focuses on near-distribution,

Table 3: FID to CIFAR-10 (ID) and MMC (%) for boundary samples (OOD) used by different OOD overconfidence mitigation methods, with CIFAR-10 results shown for reference.

| Dataset | FID | MMC |
|---|---|---|
| CIFAR-10 | 0.00 | 97.98 |
| CEDA | 7.25 | 87.39 |
| ACET | 7.49 | **99.98** |
| CODES | 3.18 | 88.84 |
| VOS | 5.28 | 88.51 |
| OE | 5.73 | 84.70 |
| OTIS | **2.45** | 91.29 |

high-confidence regions that previous schemes under-cover, clarifying the benefit of OT-based construction over heuristic boundary generation.

**Confidence Calibration on ID Data.** Tab. 4 reports expected calibration error (ECE) on ID test sets for the base classifier and models trained with OOD overconfidence mitigation methods. OTIS achieves the best ECE on CIFAR-10 and FMNIST and matches the best baseline on

Table 4: ECE ($\times 10^{-2}$) on ID test sets for the base classifier and models trained with OOD overconfidence mitigation methods.

| Dataset | - | CEDA | ACET | CCUs | CODEs | VOS | Ours | OE | CCUd |
|---|---|---|---|---|---|---|---|---|---|
| CIFAR-10 | 3.77 | 4.61 | 3.73 | 3.81 | 4.26 | 2.95 | **1.88** | 3.65 | 2.91 |
| CIFAR-100 | **6.08** | 11.34 | 9.70 | 12.88 | 11.45 | 8.97 | 6.91 | 9.69 | 6.16 |
| SVHN | 2.10 | 1.83 | 1.30 | 2.82 | 2.74 | 1.51 | 1.66 | **1.28** | 1.67 |
| MNIST | 0.21 | **0.14** | 0.16 | 0.37 | 0.31 | 0.24 | **0.14** | 0.34 | 0.30 |
| FMNIST | 4.00 | 3.61 | 4.12 | 4.53 | 4.65 | 3.39 | **3.26** | 4.50 | 3.32 |
| ImageNet | **2.05** | 3.07 | 2.64 | - | 4.75 | 4.26 | 2.71 | - | - |

MNIST. On CIFAR-100 and ImageNet datasets, several methods worsen ECE relative to the base model, whereas OTIS stays close to the baseline. These results indicate that OTIS reduces overconfidence on OOD inputs without degrading, and sometimes improving, ID calibration.

Table 5: Comparison of OTIS variants for mitigating OOD overconfidence with CIFAR-10 as ID. Columns (Gaussian-5, Uniform-3/7/5) follow the format base distribution - number of encoder/decoder layers. We report ID MMC ($\uparrow$), TE ($\downarrow$), OOD MMC ($\downarrow$), and FPR95 ($\downarrow$).

| Dataset | Gaussian-5 | | Uniform-3 | | Uniform-7 | | Uniform-5 (Ours) | |
|---|---|---|---|---|---|---|---|---|
| | ID MMC | TE | ID MMC | TE | ID MMC | TE | ID MMC | TE |
| CIFAR-10 | **96.48** | **7.31** | 96.00 | 7.60 | 92.31 | 8.55 | 95.46 | 7.52 |
| | OOD MMC | FPR95 | OOD MMC | FPR95 | OOD MMC | FPR95 | OOD MMC | FPR95 |
| SVHN | 17.77 | 1.54 | 24.95 | 3.11 | 14.27 | 1.23 | **13.18** | **1.21** |
| CIFAR-100 | 68.81 | 64.78 | 67.74 | 65.34 | **59.83** | 68.63 | 64.79 | **63.38** |
| LSUN_CR | 39.33 | 15.83 | 34.47 | 19.27 | 34.72 | 22.77 | **30.36** | **10.95** |
| Textures_C | 57.77 | 45.64 | 56.88 | 46.22 | 54.57 | 50.07 | **48.75** | **36.70** |
| Noise | 24.23 | **0.00** | 28.70 | 0.23 | 21.59 | 4.54 | **16.18** | **0.00** |
| Uniform | **10.00** | **0.00** | **10.00** | **0.00** | 10.04 | **0.00** | **10.00** | **0.00** |
| Adv. Noise | 27.91 | 3.43 | 44.51 | 3.00 | 31.37 | 1.09 | **26.42** | **0.66** |
| Adv. Samples | 59.96 | 76.78 | 60.21 | 77.55 | 59.63 | 78.64 | **57.71** | **74.71** |

**Effect of Base Distribution.** Tab. 5 compares a Gaussian base distribution (Gaussian-5) and the uniform base distribution (Uniform-5, Ours) under the same 5-layer symmetric encoder–decoder. Both yield similar CIFAR-10 test error and ID MMC, while Uniform-5 consistently achieves better OOD metrics (lower OOD MMC and FPR95). This suggests that OTIS is insensitive to the exact base distribution and that a uniform base is a strong default choice in practice.

**Effect of Autoencoder Depth.** Tab. 5 further compares OTIS with different AE depths under a uniform base (Uniform-3, Uniform-5, Uniform-7). CIFAR-10 test error and ID MMC remain close across depths, whereas Uniform-5 generally offers the best OOD trade-off, with the lowest or near-lowest OOD MMC and FPR95 on most datasets. This indicates that OTIS is robust to the AE depth and that a 5-layer symmetric encoder–decoder provides a good balance between computational cost and the effectiveness in alleviating OOD overconfidence.

## 5 RELATED WORK

**Handling OOD Overconfidence.** Test-time detection methods estimate prediction confidence via softmax responses (Hendrycks & Gimpel, 2017), temperature scaling (Liang et al., 2018), energy scores (Liu et al., 2020), or feature-space distances (Lee et al., 2018b). Though widely used, these post hoc techniques do not resolve the model's inherent overconfidence on unfamiliar inputs. A proactive alternative introduces proxy OOD inputs during training. Some methods use generative models to estimate ID boundaries and penalize confidence in low-density areas (Lee et al., 2018a; Meinke & Hein, 2020), though these often require careful tuning or partial OOD access (Li & Vasconcelos, 2020). Others employ unrelated datasets (Hendrycks et al., 2018), or apply input transformations—e.g., noise, permutation, mixing (Hein et al., 2019; Maaten & Hinton, 2008), or sample from low-likelihood regions (Du et al., 2022). These methods typically rely on heuristics and lack principled ways to characterize semantic ambiguity or boundary instability. We instead exploit semi-discrete OT geometry to identify transport-induced singularities, enabling grounded OOD example construction for confidence regularization.

**Optimal Transport (OT).** OT offers a principled framework for comparing probability measures via cost-optimal mappings (Villani et al., 2008; Santambrogio, 2015; Mérigot, 2011). It has been broadly applied in domain adaptation (Courty et al., 2017; Damodaran et al., 2018), generative modeling (Genevay et al., 2018; Tolstikhin et al., 2018), and representation learning (Cherian & Aeron, 2020; Chen et al., 2022). In contrast, we are the first to exploit structural singularities in semi-discrete OT for OOD robustness, introducing OT as a novel tool for confidence calibration.

## 6 CONCLUSION

In this paper, we have presented a theoretical perspective on OOD overconfidence by linking it to singularities in semi-discrete optimal transport, which highlight structurally ambiguous regions where classifiers tend to be overconfident. These regions provide natural targets for confidence regularization. Extensive experiments across various architectures and ID/OOD settings show that our method reduces overconfidence without sacrificing ID accuracy. We hope this work motivates further studies on robustness beyond OOD challenges.

ACKNOWLEDGEMENTS

This work was supported in part by the National Natural Science Foundation of China (62472117, 62572400, U2436208, 62372129), the Guangdong Basic and Applied Basic Research Foundation (2025A1515010157, 2024A1515012064), the Science and Technology Projects in Guangzhou (2025A03J0137, 2024B0101010002), the CCF-NetEase ThunderFire Innovation Research Funding (CCF-Netease 202514), and the Project of Guangdong Key Laboratory of Industrial Control System Security (2024B1212020010).

ETHICS STATEMENT

This work adheres to the ICLR Code of Ethics. It does not involve human subjects, sensitive user data, or potentially harmful applications. Our method is designed to improve the robustness and trustworthiness of deep neural networks against out-of-distribution inputs, and does not promote or enable misuse. All datasets used are publicly available and commonly adopted in the literature, and we have ensured no bias or discriminatory behavior is introduced during training or evaluation.

REPRODUCIBILITY STATEMENT

To facilitate reproducibility, we provide key hyper-parameters, training settings, and evaluation protocols in Sec. 4. The complete source code and reproduction instructions will be released publicly upon acceptance of this paper.

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

## A   APPENDIX

### A.1   MORE QUANTITATIVE RESULTS

**Comparison on Mitigating OOD Overconfidence Measured by FPR95.**   Tab. 6 reports the FPR95 across a broad set of ID and OOD dataset pairs. Our method consistently achieves competitive or superior performance across most settings. Compared to baseline models, which often yield FPR95 above 60%, our approach substantially reduces overconfidence, particularly on near-OOD and corrupted variants. Among methods without auxiliary data, our method significantly outperforms CEDA, ACET, CODEs, and VOS on most datasets, especially on Adversarial Noise and Adversarial Samples. For instance, on CIFAR-10 with Adversarial Noise, our method achieves 0.66% FPR95, far below VOS (94.12%) and CODEs (19.55%). Compared to methods using auxiliary datasets, such as OE and CCUd, our method is competitive while remaining free of extra supervision. These results confirm that our OT-based framework effectively identifies structurally ambiguous regions and suppresses overconfident predictions on a diverse range of OOD inputs.

**Comparison on Mitigating OOD Overconfidence Measured by AUROC.**   To further evaluate the model's ability to suppress overconfident predictions, we report the area Under the receiver operating characteristic curve (AUROC), where we use the confidence as a threshold for the detection problem (ID vs. OOD). As shown in Tab. 7, our method achieves consistently high AUROC scores across various OOD types. It outperforms all baselines on CIFAR-10 with SVHN and Noise as OOD (99.79% and 99.97%), and maintains strong performance on challenging shifts such as Adversarial Noise (100.00% on CIFAR-100). On simpler datasets like MNIST and FMNIST, it achieves near-perfect AUROC under multiple shifts. These results demonstrate the effectiveness of our method in mitigating OOD overconfidence across diverse conditions.

**Robustness to Common Corruptions.**   We treat CIFAR-10 and CIFAR-100 as ID data and their corrupted counterparts CIFAR-10-C and CIFAR-100-C as OOD inputs, modeling more realistic, corruption-based distribution shifts. Tab. 8 summarizes the results on these corruption benchmarks. On CIFAR-10-C, OTIS achieves the lowest OOD MMC and FPR95 and the highest AUROC among all methods. On CIFAR-100-C, OTIS again attains the lowest MMC and FPR95, while its AUROC remains competitive with the strongest baseline CCUd. These results show that OTIS effectively mitigates overconfidence and improves OOD detection under challenging corruption-induced shifts.

### A.2   MORE VISUALIZATION RESULTS

**Visualization of Confidence Distributions.**   We visualize the distribution of maximum softmax confidence on various OOD inputs before and after applying our method. As shown in Fig. 7, standard models exhibit a long tail of overconfident predictions even on clearly OOD samples. Our method effectively reduces high-confidence spikes and flattens the overall distribution. Fig. 8 highlights the robustness of our approach across different types of synthetic OOD inputs, including random noise and adversarial variants. In all settings, confidence mass is shifted away from the high end, indicating more calibrated predictions. Finally, Fig. 9 demonstrates that our method remains effective under large-scale settings. Despite the difficulty of the ImageNet task, our approach reduces confidence saturation on OOD datasets such as OpenImage-O and SUN, suggesting improved generalization to unseen distributions.

**Visualization of Transport-Induced OOD Samples (OTIS).**   We visualize OTIS instances generated from different datasets in Figs. 10–12. Across all domains, OTIS instances exhibit seman-

Table 6: Comparison of eight methods for mitigating OOD overconfidence, measured by FPR95 (%, ↓). Note: OE and CCUd leverage auxiliary datasets, while the others do not.

| ID | OOD | Baseline | CEDA | ACET | CCUs | CODEs | VOS | Ours | OE | CCUd |
|---|---|---|---|---|---|---|---|---|---|---|
| with auxiliary dataset | | | | | | | | | ✓ | ✓ |
| CIFAR-10 | SVHN | 59.51 | 55.78 | 69.99 | 21.20 | 72.23 | 47.5 | **1.21** | 31.87 | 8.24 |
| | CIFAR-100 | 64.99 | 70.69 | 69.66 | **40.18** | 74.32 | 62.93 | 63.38 | 66.90 | 61.18 |
| | LSUN_CR | 45.27 | 35.11 | 39.40 | 25.27 | 35.25 | 42.33 | **10.95** | 45.67 | 12.02 |
| | Textures_C | 66.49 | 57.46 | 63.76 | 25.32 | 51.54 | 57.77 | 36.70 | 50.55 | **20.05** |
| | Noise | 59.49 | 17.80 | 25.27 | 32.75 | 18.06 | 11.00 | **0.00** | **0.00** | 0.01 |
| | Uniform | 68.59 | **0.00** | **0.00** | **0.00** | **0.00** | 62.95 | **0.00** | 0.39 | **0.00** |
| | Adv. Noise | 97.46 | 31.94 | **0.00** | **0.00** | 19.55 | 94.12 | 0.66 | 84.61 | **0.00** |
| | Adv. Samples | 94.28 | 92.01 | 76.84 | 100.00 | 91.74 | 92.81 | **74.71** | 86.64 | 99.83 |
| CIFAR-100 | SVHN | 71.39 | 86.46 | 79.01 | 17.90 | 82.08 | 75.07 | **8.81** | 66.86 | 10.00 |
| | CIFAR-10 | 79.49 | 86.02 | 83.51 | 12.92 | 82.42 | 79.76 | 77.38 | 82.43 | **10.07** |
| | LSUN_CR | 78.60 | 72.86 | 64.81 | 9.93 | 45.45 | 76.48 | 55.82 | 64.31 | **9.01** |
| | Textures_C | 80.11 | 89.29 | 84.52 | 83.16 | 75.30 | 78.74 | 75.57 | 75.62 | **74.47** |
| | Noise | 99.61 | 99.75 | 99.87 | 33.23 | 97.83 | 99.49 | 28.20 | **0.00** | 0.55 |
| | Uniform | 97.95 | **0.00** | **0.00** | **0.00** | **0.00** | 98.78 | **0.00** | **0.00** | **0.00** |
| | Adv. Noise | 99.73 | 21.48 | **0.00** | **0.00** | 27.31 | 99.34 | 7.41 | 84.69 | **0.00** |
| | Adv. Samples | 92.89 | 92.57 | 85.65 | 100.00 | 90.60 | 88.84 | 86.09 | **82.74** | 99.00 |
| SVHN | CIFAR-10 | 30.45 | 45.61 | 18.12 | 16.47 | **7.68** | 27.09 | 11.45 | 19.85 | 8.27 |
| | CIFAR-100 | 30.61 | 46.32 | 19.84 | 17.45 | 17.64 | 28.89 | **13.29** | 21.00 | 16.51 |
| | LSUN_CR | 17.50 | 31.89 | 10.01 | 29.24 | 49.03 | 15.75 | 11.04 | 10.87 | **10.00** |
| | Textures_C | 28.62 | 46.15 | 12.61 | 10.90 | 8.63 | 26.79 | 41.17 | 11.40 | **0.00** |
| | Noise | 99.54 | 95.38 | 96.19 | 97.26 | 99.95 | 96.23 | **90.12** | 95.48 | 96.28 |
| | Uniform | 28.91 | **0.00** | **0.00** | **0.00** | **0.00** | 26.65 | **0.00** | **0.00** | **0.00** |
| | Adv. Noise | 77.50 | 60.74 | **0.00** | **0.00** | 11.47 | 74.55 | 7.98 | 4.60 | **0.00** |
| | Adv. Samples | 66.55 | 67.06 | 46.26 | 99.67 | 39.60 | 63.15 | 59.07 | **23.14** | 99.00 |
| MNIST | FMNIST | 1.44 | 1.56 | 0.48 | 6.78 | 0.80 | 2.48 | 0.36 | 0.85 | **0.26** |
| | EMNIST | 22.15 | 21.56 | 19.88 | 40.95 | 19.70 | 21.78 | **15.38** | 21.60 | 35.10 |
| | GrayCIFAR | 0.03 | **0.00** | **0.00** | **0.00** | **0.00** | 0.04 | 0.30 | **0.00** | **0.00** |
| | Kylberg | **0.00** | **0.00** | **0.00** | **0.00** | **0.00** | **0.00** | **0.00** | **0.00** | **0.00** |
| | Noise | **0.00** | **0.00** | **0.00** | **0.00** | **0.00** | **0.00** | **0.00** | **0.00** | **0.00** |
| | Uniform | **0.00** | **0.00** | **0.00** | **0.00** | **0.00** | **0.00** | **0.00** | **0.00** | **0.00** |
| | Adv. Noise | 0.04 | **0.00** | **0.00** | **0.00** | **0.00** | **0.00** | **0.00** | **0.00** | **0.00** |
| | Adv. Samples | 4.06 | 0.43 | **0.00** | **0.00** | 1.44 | 0.99 | **0.00** | **0.00** | 0.17 |
| FMNIST | MNIST | 59.07 | 48.92 | **35.12** | 57.83 | 79.43 | 51.54 | 53.38 | 40.78 | 52.88 |
| | EMNIST | 66.61 | 52.37 | 43.71 | 42.97 | 81.04 | 53.36 | **41.62** | 47.97 | 43.94 |
| | GrayCIFAR | 86.65 | 22.53 | 16.56 | 11.90 | 7.61 | 66.91 | 7.46 | **0.47** | 7.10 |
| | Kylberg | 92.75 | 1.07 | 6.03 | 10.85 | 0.31 | 65.2 | 2.56 | 0.16 | **0.00** |
| | Noise | 92.99 | 0.02 | 21.32 | 2.42 | **0.00** | 16.77 | **0.00** | **0.00** | 0.01 |
| | Uniform | 79.85 | **0.00** | **0.00** | **0.00** | **0.00** | 26.54 | **0.00** | 0.02 | **0.00** |
| | Adv. Noise | 99.92 | 1.30 | **0.00** | **0.00** | 0.91 | 99.26 | **0.00** | 95.78 | **0.00** |
| | Adv. Samples | 82.28 | 28.65 | 17.19 | 100.00 | 16.65 | 64.45 | 38.15 | **12.51** | 98.50 |

tically coherent yet structurally ambiguous characteristics that differ from typical ID patterns. On ImageNet (Fig. 10), the generated instances contain mixed or blurred attributes resembling multiple classes (e.g., dog–cat or ship–truck), often with disrupted contours and overlaid textures. On CIFAR-10 (Fig. 11), OTIS instances present hybrid features and indistinct boundaries, lacking the sharp discriminative traits of clean samples. On FMNIST (Fig. 12), shape distortions and pattern mixing are especially prominent, producing examples that lie between known fashion categories. These results demonstrate that our OT-based sampling process successfully identifies regions of structural uncertainty and generates visually plausible OOD inputs for suppressing overconfidence.

## A.3 STATEMENT ON LLM USAGE

LLMs were used solely as a writing-assistance tool to polish the language of this manuscript (e.g., grammar, phrasing, and clarity). They were not involved in research ideation, experiment design, data analysis, or substantive content generation.

Table 7: Comparison of eight methods for mitigating OOD overconfidence, measured by AUROC (%, ↑). Note: OE and CCUd leverage auxiliary datasets, while the others do not.

| ID | OOD | Baseline | CEDA | ACET | CCUs | CODEs | VOS | Ours | OE | CCUd |
|---|---|---|---|---|---|---|---|---|---|---|
| with auxiliary dataset | | | | | | | | | ✓ | ✓ |
| CIFAR-10 | SVHN | 91.29 | 92.56 | 89.00 | 88.75 | 86.67 | 92.56 | **99.79** | 95.27 | 85.42 |
| | CIFAR-100 | **88.35** | 85.69 | 85.64 | 85.83 | 83.77 | 85.76 | 87.19 | 86.15 | 82.38 |
| | LSUN_CR | 93.80 | 95.42 | 94.93 | 90.66 | 94.57 | 92.94 | 98.26 | 94.12 | **98.34** |
| | Textures_C | 88.50 | 90.26 | 88.31 | 94.52 | 90.10 | 88.32 | 93.01 | 90.04 | **97.29** |
| | Noise | 93.60 | 96.40 | 95.47 | 98.38 | 97.71 | 98.14 | **99.97** | 99.96 | 97.22 |
| | Uniform | 90.46 | **100.00** | **100.00** | **100.00** | **100.00** | 91.61 | **100.00** | 99.84 | **100.00** |
| | Adv. Noise | 56.38 | 85.36 | **100.00** | **100.00** | 93.11 | 92.96 | 95.78 | 98.77 | **100.00** |
| | Adv. Samples | 54.77 | 57.31 | 66.79 | 63.40 | 65.68 | 59.26 | 67.27 | 63.11 | 62.79 |
| CIFAR-100 | SVHN | 84.15 | 82.14 | 86.35 | 86.10 | 89.82 | 90.6 | **98.04** | 93.37 | 97.37 |
| | CIFAR-10 | 78.61 | 72.33 | 74.20 | 67.54 | 76.23 | 77.83 | 74.05 | 74.61 | **80.25** |
| | LSUN_CR | 80.38 | 81.45 | 85.16 | 86.08 | **89.83** | 81.89 | 87.28 | 85.55 | 82.32 |
| | Textures_C | 77.74 | 68.02 | 72.09 | **86.24** | 78.45 | 78.09 | 79.00 | 75.45 | 76.45 |
| | Noise | 53.61 | 52.09 | 60.34 | 53.55 | 69.60 | 54.27 | 61.93 | **100.00** | 56.60 |
| | Uniform | 69.77 | **100.00** | **100.00** | 99.99 | **100.00** | 75.63 | **100.00** | 99.99 | **100.00** |
| | Adv. Noise | 52.33 | 91.12 | **100.00** | **100.00** | 90.83 | 69.57 | **100.00** | 73.15 | **100.00** |
| | Adv. Samples | 60.26 | 74.23 | 84.75 | 82.88 | 81.20 | 84.06 | 85.36 | **89.11** | 85.59 |
| SVHN | CIFAR-10 | 91.68 | 92.26 | 96.26 | 96.26 | 98.45 | 92.44 | 89.83 | 95.94 | **99.94** |
| | CIFAR-100 | 91.82 | 92.09 | 96.01 | 97.96 | 97.64 | 91.91 | 98.18 | 95.66 | **99.92** |
| | LSUN_CR | 96.04 | 94.87 | 97.80 | 95.15 | 92.43 | 95.75 | 95.47 | 97.59 | **99.90** |
| | Textures_C | 92.93 | 90.84 | 97.36 | 99.24 | 98.33 | 92.52 | 97.56 | 97.68 | **99.94** |
| | Noise | 84.09 | 87.07 | 88.07 | 96.18 | 91.83 | 87.35 | 95.72 | **99.99** | 95.07 |
| | Uniform | 94.22 | **100.00** | **100.00** | **100.00** | **100.00** | 93.19 | **100.00** | **100.00** | **100.00** |
| | Adv. Noise | 50.25 | 76.98 | **100.00** | **100.00** | 95.62 | 85.05 | 92.43 | 98.64 | **100.00** |
| | Adv. Samples | 61.44 | 76.69 | 81.43 | 84.84 | 89.10 | 74.44 | 86.68 | 92.06 | **92.79** |
| MNIST | FMNIST | 99.34 | 99.19 | 99.69 | 99.25 | 99.46 | 99.11 | **99.77** | 99.50 | 99.61 |
| | EMNIST | 94.74 | 94.74 | 95.31 | 94.05 | 95.06 | 94.9 | **96.55** | 94.84 | 96.15 |
| | GrayCIFAR | 99.83 | 99.94 | **100.00** | 99.99 | 99.95 | 99.85 | 99.87 | 99.99 | 99.99 |
| | Kylberg | 99.99 | **100.00** | **100.00** | **100.00** | **100.00** | **100.00** | **100.00** | **100.00** | **100.00** |
| | Noise | **100.00** | **100.00** | **100.00** | 99.99 | **100.00** | **100.00** | **100.00** | **100.00** | 99.99 |
| | Uniform | **100.00** | **100.00** | **100.00** | **100.00** | **100.00** | **100.00** | **100.00** | **100.00** | **100.00** |
| | Adv. Noise | 97.55 | 99.28 | **100.00** | **100.00** | 99.57 | **100.00** | **100.00** | 98.91 | **100.00** |
| | Adv. Samples | 97.80 | 98.32 | 99.75 | **100.00** | 97.98 | 98.08 | **100.00** | 98.88 | 99.99 |
| FMNIST | MNIST | 90.72 | 94.16 | 95.58 | 85.08 | 79.23 | 93.16 | **97.87** | 95.10 | 97.26 |
| | EMNIST | 87.70 | 92.71 | 94.02 | 88.12 | 78.44 | 92.37 | 92.96 | 93.40 | **96.11** |
| | GrayCIFAR | 83.89 | 97.23 | 97.89 | 99.00 | 98.97 | 92.25 | 98.44 | **99.90** | 98.24 |
| | Kylberg | 79.93 | 99.78 | 99.18 | 99.30 | **99.92** | 92.97 | 98.74 | **99.92** | 99.08 |
| | Noise | 86.65 | 99.80 | 96.46 | 97.94 | **100.00** | 97.01 | 97.96 | **100.00** | 96.98 |
| | Uniform | 90.90 | **100.00** | **100.00** | 99.49 | **100.00** | 96.54 | **100.00** | 99.83 | 99.98 |
| | Adv. Noise | 57.87 | 99.80 | **100.00** | 99.94 | 99.86 | 67.08 | 98.87 | 83.45 | 99.99 |
| | Adv. Samples | 81.54 | 96.29 | 97.57 | 82.72 | 96.85 | 86.76 | 93.84 | **97.74** | 87.60 |

Table 8: Comparison of different methods for mitigating OOD overconfidence under common corruptions, treating CIFAR-10/100 as ID data and CIFAR-10-C/100-C as OOD inputs. We report OOD MMC (↓), FPR95 (↓), and AUROC (↑), all in percent (%).

| Method | CIFAR-10-C | | | CIFAR-100-C | | |
|---|---|---|---|---|---|---|
| | MMC | FPR95 | AUROC | MMC | FPR95 | AUROC |
| Base | 93.21 | 83.30 | 70.43 | 65.61 | 84.06 | 68.92 |
| CEDA | 86.81 | 80.92 | 70.66 | 69.85 | 89.30 | 63.17 |
| ACET | 90.81 | 85.00 | 66.16 | 71.80 | 87.40 | 65.62 |
| CCUs | 85.42 | 80.96 | 66.16 | 66.19 | 79.78 | 65.94 |
| CODES | 84.44 | 84.80 | 68.00 | 74.96 | 85.80 | 67.04 |
| VOS | 90.89 | 82.38 | 66.07 | 70.90 | 84.72 | 67.67 |
| Ours | **70.78** | **66.16** | **76.63** | **59.22** | **73.82** | 72.72 |
| OE | 87.75 | 79.86 | 70.15 | 71.18 | 85.20 | 66.16 |
| CCUd | 83.82 | 68.79 | 73.99 | 63.27 | 78.16 | **79.20** |

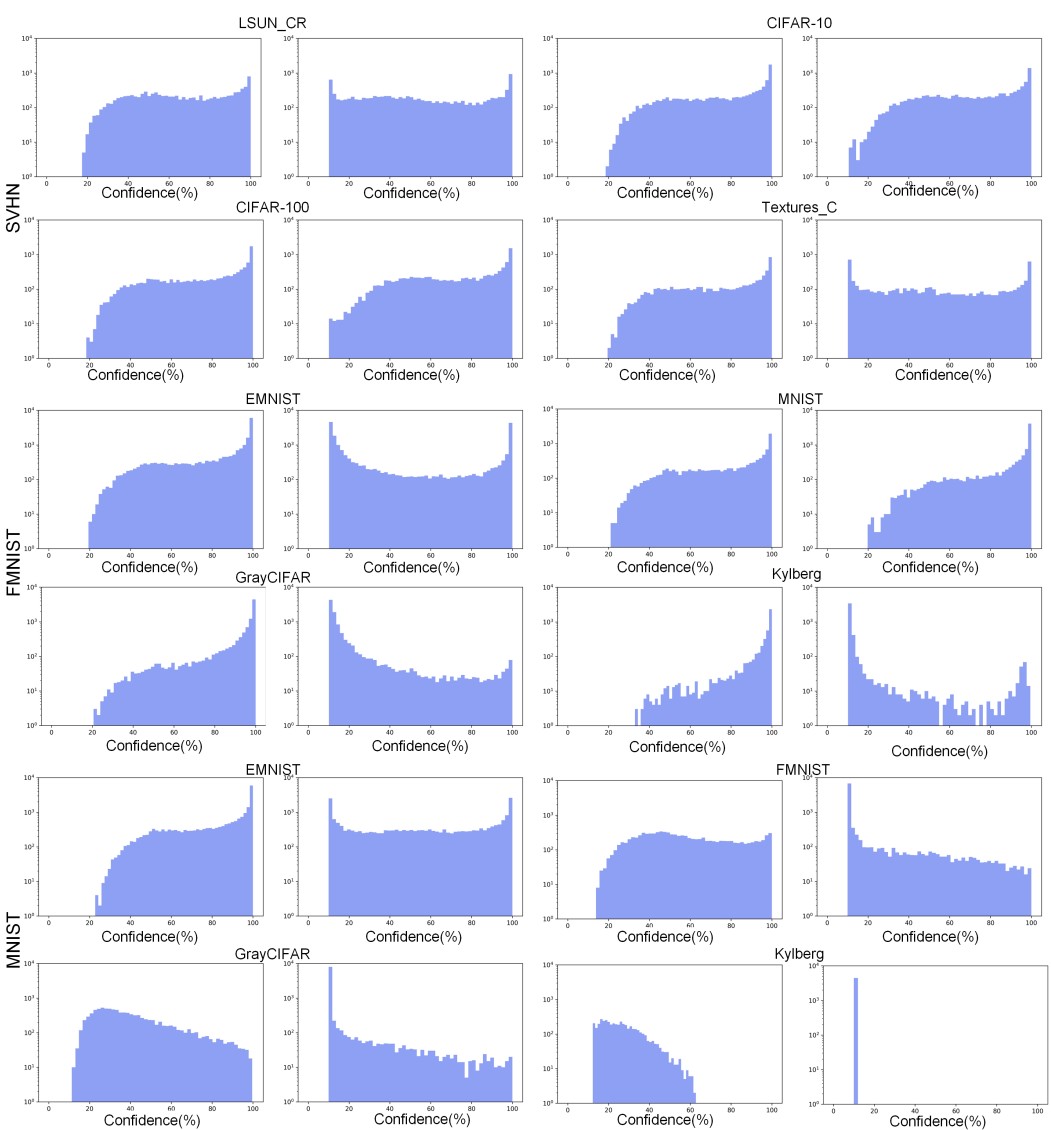

Figure 7: Histograms of maximum softmax confidence on OOD samples from common datasets. Each model is trained on SVHN, FMNIST, or MNIST using ResNet-18, and tested on datasets such as CIFAR-100, LSUN, and EMNIST.

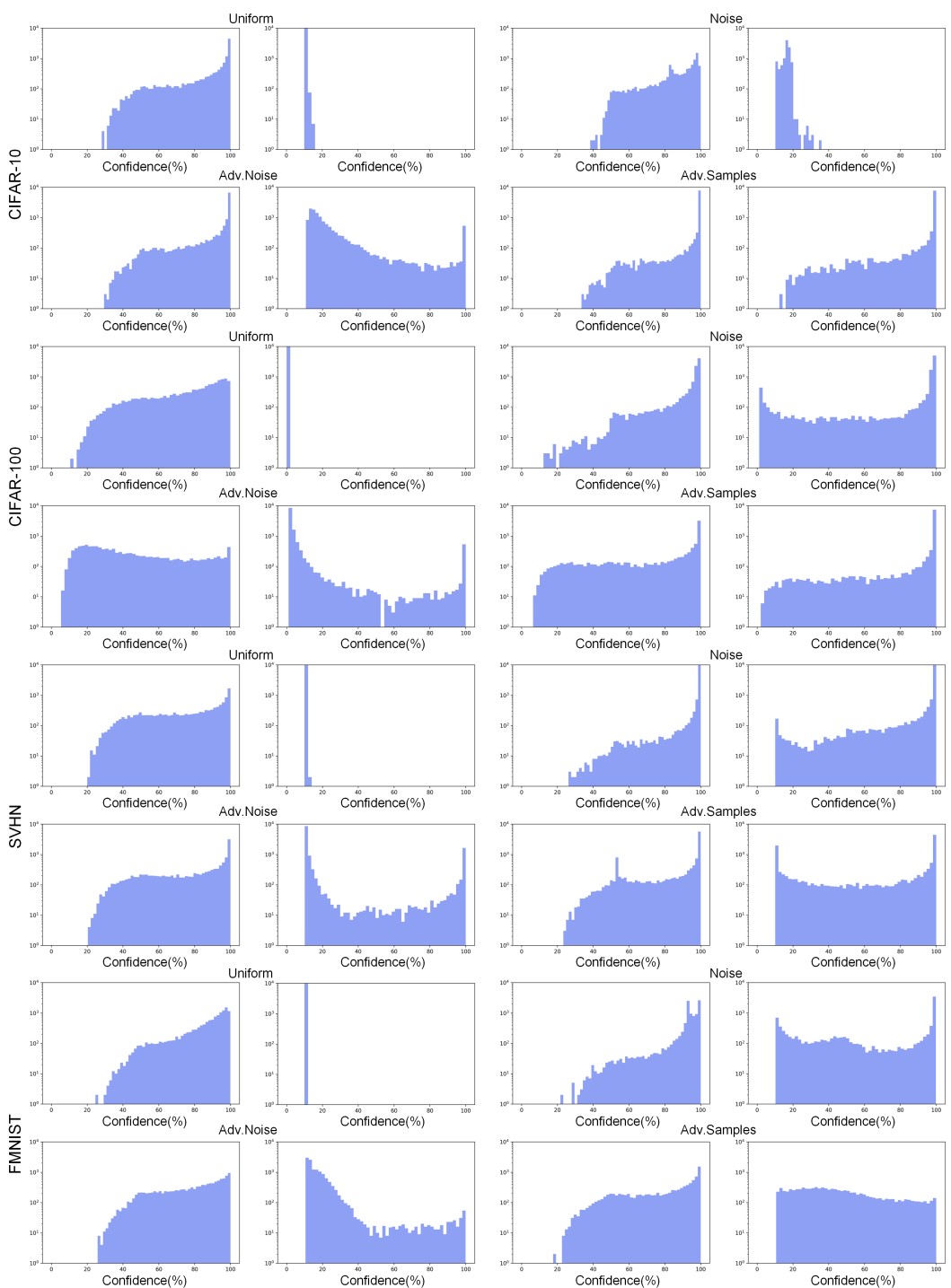

Figure 8: Histograms of confidence on synthetic OOD inputs. Each model is trained on CIFAR-10, CIFAR-100, SVHN, or FMNIST using ResNet-18, and evaluated on Uniform, Noise, Adversarial Noise, and Adversarial Samples.

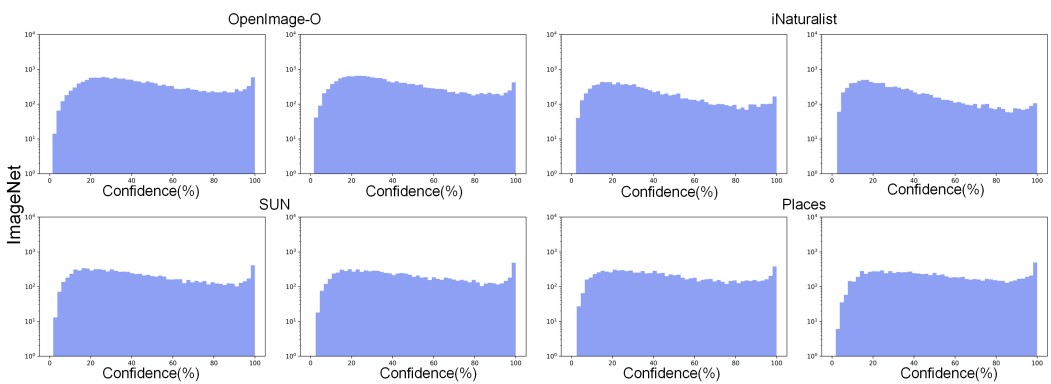

Figure 9: Histograms of maximum confidence scores on large-scale OOD datasets. A ResNet-50 is trained on ImageNet and evaluated on OpenImage-O, iNaturalist, SUN, and Places.

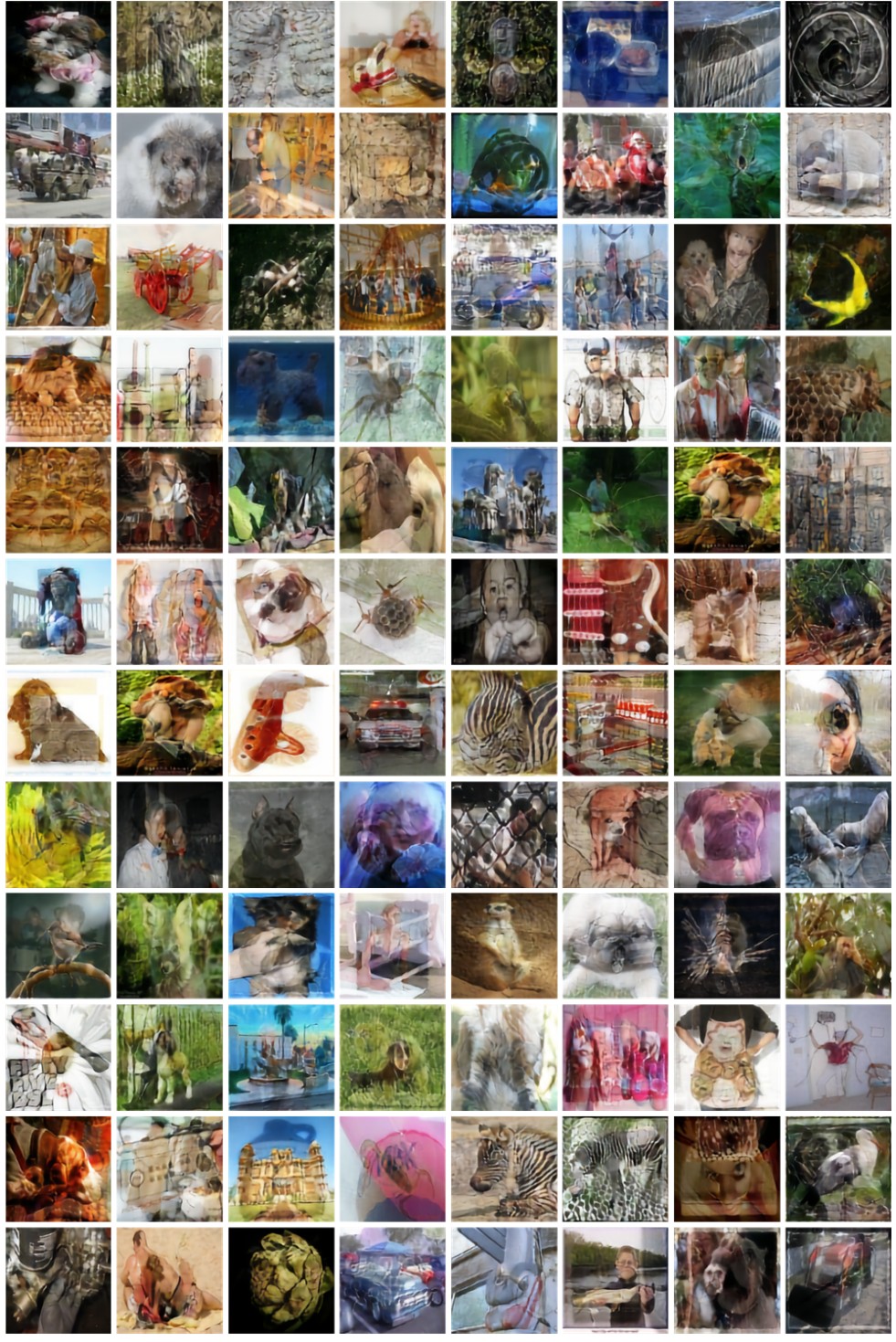

Figure 10: Visualizations of OTIS generated from ImageNet.

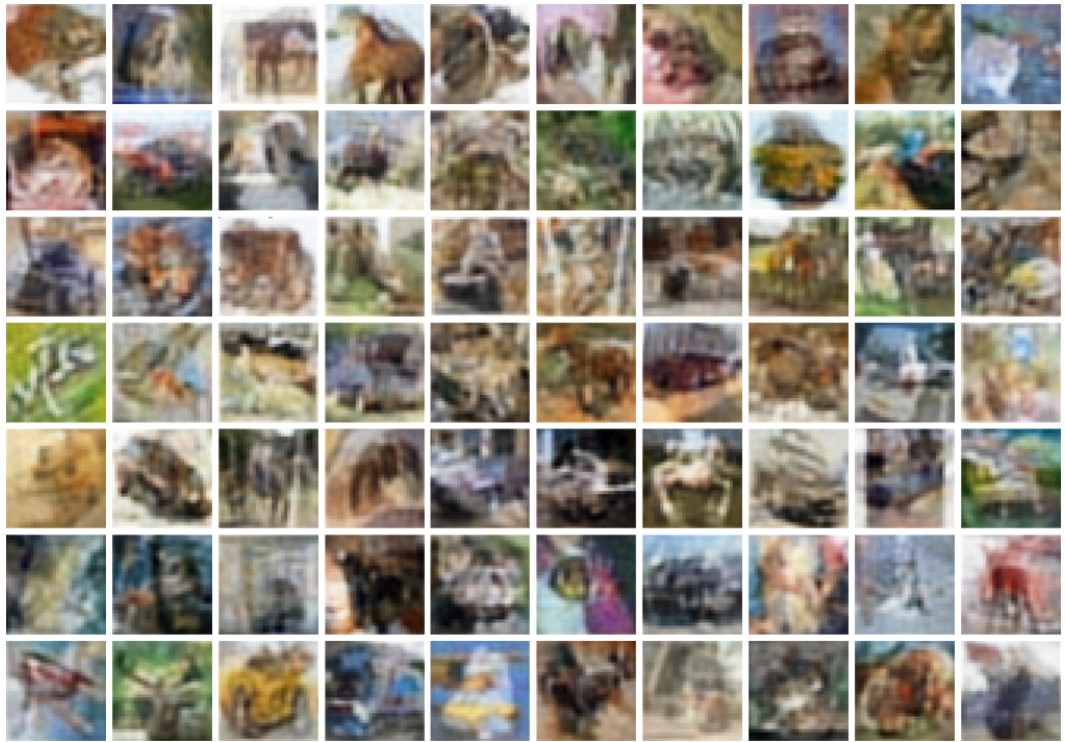

Figure 11: Visualizations of OTIS generated from CIFAR-10.

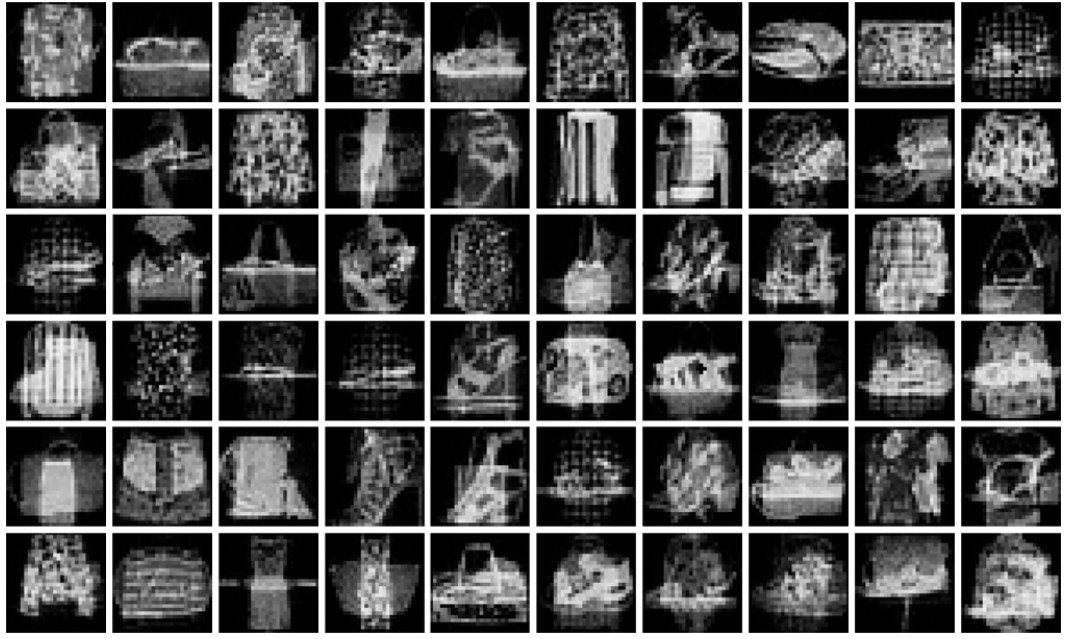

Figure 12: Visualizations of OTIS generated from FMNIST.

