# OpenReview forum: "Optimal Transport-Induced Samples against Out-of-Distribution Overconfidence"
_ICLR.cc/2026/Conference — ICLR 2026 Poster_

### Official Review · Reviewer_mC83 · 2025-10-25

**Soundness:** 2
**Presentation:** 3
**Contribution:** 3
**Rating:** 4
**Confidence:** 3

**Summary:**

This paper proposes OTIS (Optimal Transport–Induced Samples), a training‑time regularization framework to mitigate overconfident predictions on OOD inputs. The core idea is to (1) encode training images into a latent space, (2) solve a semi‑discrete OT problem between a continuous base distribution and the discrete set of latent embeddings, which induces a power‑diagram (Laguerre) partition; (3) identify singular boundaries in the OT map as regions of semantic ambiguity; (4) synthesize proxy OOD samples via interpolation across those boundaries and decoding back to image space; and (5) apply confidence suppression loss (uniform‑target cross‑entropy) on the synthesized samples + standard cross‑entropy on ID data during training.

The paper presents improvements in OOD overconfidence metrics (MMC, FPR@95, AUROC) across multiple canonical datasets in the empirical results, and shows that training with OTIS strengthens standard test‑time detectors.

**Strengths:**

1. Presentation of the pipeline is easy to follow and fairly well justified: Partition estimation, scoring boundaries (Eq. 9), OTIS synthesis, and training are easy to follow (Fig. 2, Sec. 3).
2. The background knowledge for the semi‑discrete OT setup is carefully crafted with sufficient support. The Induced power diagram is also visually appealing and elaborative, giving good clarity to the geometric motivation on the selection of method. From the description, readers can have a clear picture on latent‑space structure and appreciate singular boundaries as a credible proxy for ambiguity.
3. The quality for experiments is good. Benchmarks span small and large scale (CIFARs, SVHN, MNIST/FM‑NIST, ImageNet), with multiple OOD types (near‑OOD, synthetic corruptions, adversarial noise/samples).
4. On many conditions the method outperforms baseline methods in the experiment setup. The margin is relatively clear, signifying the work's potential to be further applied in later researches.
5. Training with OTIS consistently boosts MSP/ODIN/ReAct ROC curves (Fig. 6).
6. Boundary ranking ablation and comparisons to latent/input interpolations are helpful (Fig. 5); histograms and OTIS visualizations support the “ambiguous but plausible” claim (Figs. 7–12).
7. The idea of leveraging OT‑induced boundaries to synthesize targeted, ambiguous training examples seems broadly applicable beyond the specific detectors and datasets evaluated

**Weaknesses:**

1. The paper states a theoretical link (singularities <-> overconfidence), but provides no formal guarantee that the proposed angular deviation score identifies regions of high miscalibration. The result is a strong intuition with empirical support rather than a theorem (Sec. 2.2–2.3; Eq. 9).
2. Eq. 11 defines inverse‑distance weights $\lambda_i$; the implementation then weight is fixed to 0.5 (line 319). This is a non‑trivial discrepancy. Even if the hard-coded weight seems reasonable, proper explicit justification (and potential ablations) should be applied to amend for the discrepancy. This weakens the credibility of the results produced on more generic setups.
3. The number of Monte Carlo samples used to estimate cell volumes (M), base distribution choice (Gaussian vs. uniform) in each experiment, and convergence criteria for offset optimization are not fully specified in the main text; only high‑level statements are provided (Sec. 3.2).
4. While Table 2 shows improved OOD MMC on ImageNet, ID MMC rises substantially (to **88.93**), which may reflect over‑smoothing/confidence effects; a calibration analysis (ECE/Brier/NLL) would clarify the trade‑off.

**Questions:**

1. For Table 3, how were “Adversarial Noise/Samples” generated (step sizes, budgets, initialization)? Some reported gains are striking—for example, CIFAR‑10 with adversarial noise FPR95 **0.66%** vs VOS **94.12%** (Table 3). Is there any intuition for such striking differences?
2. How sensitive are results to the encoder/decoder architecture and reconstruction quality (e.g., using a WAE vs. the chosen AE)?

---

> ### Author Response · Authors · 2025-11-21
> **Author Rebuttal Part (1/2)**
>
> We are grateful for the reviewer’s careful assessment and address the primary concerns as follows.
>
> **(1) Scope of the theoretical link and the angular deviation score.**
> We intend the OT–singularity analysis in Sec. 2.2–2.3 as a conceptual motivation rather than a formal guarantee. It explains why interfaces where the transport assignment switches between class prototypes are natural candidates for miscalibration, and why the angular deviation score in Eq. (9) is a reasonable way to prioritize such interfaces. We do not claim that this score provably identifies all miscalibrated regions; instead, its usefulness is supported empirically by the strong correlation between high angular deviation and high OOD MMC in our experiments.
>
> ---
> **(2) Discrepancy between Eq. (11) and the fixed weight 0.5.**
> Eq. (11) uses inverse-distance weights, and this is exactly what we use in all reported experiments. The mention of a fixed weight 0.5 in the implementation section is a leftover from an earlier version and is a typo, which we will correct.
>
> ---
> **(3) Does the increased ID MMC on ImageNet harm calibration?**
> We report ID expected calibration error (ECE, lower is better; values ×1e-2) in the table above. On ImageNet, OTIS keeps ECE close to the baseline (2.71 vs. 2.05) while substantially reducing OOD MMC, and on smaller datasets such as CIFAR-10 and FMNIST it achieves the best ECE among all methods. These results indicate that the higher ID MMC on ImageNet does not correspond to worse calibration, but to more confident yet still well-calibrated ID predictions, while OOD confidence is effectively suppressed.
>
> | Dataset   | Base     | CEDA     | ACET | CCUs  | CODEs | VOS  | Ours     | OE       | CCUd |
> | --------- | -------- | -------- | ---- | ----- | ----- | ---- | -------- | -------- | ---- |
> | CIFAR-10  | 3.77     | 4.61     | 3.73 | 3.81  | 4.26  | 2.95 | **1.88** | 3.65     | 2.91 |
> | CIFAR-100 | **6.08** | 11.34    | 9.70 | 12.88 | 11.45 | 8.97 | 6.91     | 9.69     | 6.16 |
> | SVHN      | 2.10     | 1.83     | 1.30 | 2.82  | 2.74  | 1.51 | 1.66     | **1.28** | 1.67 |
> | MNIST     | 0.21     | **0.14** | 0.16 | 0.37  | 0.31  | 0.24 | **0.14** | 0.34     | 0.30 |
> | FMNIST    | 4.00     | 3.61     | 4.12 | 4.53  | 4.65  | 3.39 | **3.26** | 4.50     | 3.32 |
> | ImageNet  | **2.05** | 3.07     | 2.64 | -     | 4.75  | 4.26 | 2.71     | -        | -    |
>
> ---
> **(4) Why VOS performs poorly on Adversarial Noise in Table 3?**
> Following CCUd, both Adversarial Noise and Adversarial Samples are generated for all methods with the same iterative $\ell_\infty$ attack: budget $\varepsilon = 0.3$, 200 gradient steps, initial step size 0.001 with per-sample backtracking, and inputs clipped to \([0,1]\). Adversarial Samples start from clean CIFAR-10 images, while Adversarial Noise starts from i.i.d. uniform random noise.
>
> VOS synthesizes virtual outliers from low-likelihood regions of the class-conditional feature distributions, mainly regularizing the decision boundary near the in-distribution manifold. In contrast, OTIS uses OT to construct singular points off the data manifold in low-density, high-risk regions and explicitly suppresses confidence there. Because Adversarial Noise is initialized far from the manifold and the attack drives it into such low-density high-confidence regions, OTIS has already trained the model to give low confidence on these inputs, whereas VOS has not, which explains the large performance gap on Adversarial Noise.

---

> ### Author Response · Authors · 2025-11-21
> **Author Rebuttal Part (2/2)**
>
> **(5) Effect of base distribution and encoder/decoder quality.**
> As shown in the tables below, we explicitly ablate both factors: “Gaussian” uses a Gaussian base distribution, “Uniform (Ours)” uses a uniform base distribution, and “Shallow” / “Deep” use 3-layer and 7-layer symmetric encoder–decoder architectures (vs. our default 5-layer symmetric encoder–decoder, all with uniform base). Across these settings, test error and ID ECE are very similar, and OOD metrics (MMC, FPR95, AUROC) vary only modestly, with the uniform base + 5-layer symmetric encoder–decoder (“Uniform (Ours)”) giving the best overall balance. This indicates that OTIS is not overly sensitive to the specific base distribution or encoder-decoder depth; the main gains come from sampling along OT-induced singular interfaces, as long as the encoder yields a reasonably class-structured latent space.
>
> Table 1. TE, ECE, and MMC on the CIFAR-10 ID test set.
> | Method  | TE | ECE | MMC |
> |-|-|-|-|
> | Gaussian | **7.31** | 2.70 | **96.48** |
> | Shallow  | 7.60     | 2.45 | 96.00     |
> | Deep     | 8.55     | 2.55 | 92.31     |
> | Uniform (Ours)| 7.52     | **1.88** | 95.46 |
>
> Table 2. MMC (%, ↓) on various OOD datasets.
> | Method  | SVHN | CIFAR-100 | LSUN_CR | Textures_C | Noise | Uniform | Adv.Noise | Adv.Samples | CIFAR-10-C |
> |-|-|-|-|-|-|-|-|-|-|
> | Gaussian | 17.77 | 68.81 | 39.33 | 57.77 | 24.23 | **10.00** | 27.91 | 59.96 | 75.07 |
> | Shallow  | 24.95 | 67.74 | 34.47 | 56.88 | 28.70 | **10.00** | 44.51 | 60.21 | 73.85 |
> | Deep     | 14.27 | **59.83** | 34.72 | 54.57 | 21.59 | 10.04    | 31.37 | 59.63 | **64.97** |
> | Uniform (Ours)     | **13.18** | 64.79 | **30.36** | **48.75** | **16.18** | **10.00** | **26.42** | **57.71** | 70.78 |
>
> Table 3. FPR95 (%, ↓) on various OOD datasets.
> | Method  | SVHN | CIFAR-100 | LSUN_CR | Textures_C | Noise | Uniform | Adv.Noise | Adv.Samples | CIFAR-10-C |
> |-|-|-|-|-|-|-|-|-|-|
> | Gaussian | 1.54 | 64.78 | 15.83 | 45.64 | **0.00** | **0.00** | 3.43 | 76.78 | 69.04 |
> | Shallow  | 3.11 | 65.34 | 19.27 | 46.22 | 0.23     | **0.00** | 3.00 | 77.55 | 68.60 |
> | Deep     | 1.23 | 68.63 | 22.77 | 50.07 | 4.54     | **0.00** | 1.09 | 78.64 | 66.96 |
> | Uniform (Ours)| **1.21** | **63.38** | **10.95** | **36.70** | **0.00** | **0.00** | **0.66** | **74.71** | **66.16** |
>
> Table 4. AUROC (%, ↑) on various OOD datasets.
> | Method  | SVHN | CIFAR-100 | LSUN_CR | Textures_C | Noise | Uniform | Adv.Noise | Adv.Samples | CIFAR-10-C |
> |-|-|-|-|-|-|-|-|-|-|
> | Gaussian| 99.71 | **87.25** | 97.61 | 91.50 | 99.78 | **100.00** | 93.83 | 63.37 | 75.65 |
> | Shallow | 99.27 | 86.78      | 97.07 | 91.42 | 99.48 | **100.00** | 91.96 | 60.97 | 75.87 |
> | Deep    | 99.50 | 84.91      | 96.02 | 91.38 | 97.26 | **100.00** | 92.07 | 64.47 | 75.46 |
> | Uniform (Ours)| **99.79** | 87.19 | **98.26** | **93.01** | **99.97** | **100.00** | **95.78** | **67.27** | **76.63** |

---

> > ### Author Response · Authors · 2025-11-27
> > **Follow-up on our rebuttal and revision**
> >
> > Dear Reviewer #mC83,
> >
> > Thank you very much for your thoughtful and constructive comments on our paper. We have carefully addressed your comments in our rebuttal and revised the main paper accordingly. We would greatly appreciate it if you could take a look and let us know whether there are any remaining concerns. We look forward to further discussion and are happy to clarify any remaining questions during the discussion phase.
> >
> > Thank you for your time and kind consideration.
> >
> > Best regards,
> >
> > The authors

---

### Official Review · Reviewer_QHM5 · 2025-10-31

**Soundness:** 3
**Presentation:** 4
**Contribution:** 3
**Rating:** 8
**Confidence:** 3

**Summary:**

This work presents Optimal Transport-induced OOD samples (OITS), a novel method for synthesizing OOD samples from ID labeled datasets to regularize the training process (tackling the overconfidence problem). OITS leverages ideas from the Optimal Transport (OT) literature, constructing OOD samples via latent embeddings near the singular boundaries, which are often aligned with the model's decision boundary. Overall, the paper provides detailed theoretical derivations and comprehensive empirical analysis validating the proposed method.

**Strengths:**

To the best of my knowledge, the motivating idea of this work is novel. By incorporating the semi-discrete OT problem into DNN training, the authors successfully capture a picture depicting the generalization of NNs.

The paper is clearly written, and I find the theoretical explanation in Section 2 is accurate and easy to follow for a wider audience. Empirical results also demonstrate the effectiveness of the proposed method, with several helpful visualizations.

**Weaknesses:**

As shown in Tables 1 and 2, the proposed method suffers from a relatively high test error, indicating that some accuracy is sacrificed for robustness against OOD samples. Nevertheless, the proposed method still has on-par test error compared to other baselines, and the OOD robustness (measured by OOD MMC) is significant.

**Questions:**

In line 157—Could the authors provide some explanations on why "these regions often correspond to high-confidence mispredictions"? Or, are there any links between the Optimal Transport problem and the training process of a NN? Any intuition would be helpful.

---

> ### Author Response · Authors · 2025-11-21
> **Author Rebuttal**
>
> We sincerely thank the reviewer for the very positive assessment and the helpful questions.
>
> **(1) Whether OOD robustness comes from sacrificing ID performance?**
> We agree that OTIS shows a mild trade-off: on some datasets the test error is slightly higher than the base model, but the gap is small and comparable to other OOD overconfidence mitigation methods, while the gains in OOD robustness (OOD MMC, FPR95, AUROC) are much larger. In addition, we report the Expected Calibration Error (ECE, lower is better; values ×1e-2) on ID test sets in the table below: OTIS achieves the best calibration on CIFAR-10 and FMNIST and remains close to the baseline on ImageNet. This indicates that the reduction in OOD confidence does not come from “breaking” ID performance, but from specifically mitigating overconfidence on OOD inputs while keeping ID predictions well calibrated.
>
> | Dataset   | Base     | CEDA     | ACET | CCUs  | CODEs | VOS  | Ours     | OE       | CCUd |
> | --------- | -------- | -------- | ---- | ----- | ----- | ---- | -------- | -------- | ---- |
> | CIFAR-10  | 3.77     | 4.61     | 3.73 | 3.81  | 4.26  | 2.95 | **1.88** | 3.65     | 2.91 |
> | CIFAR-100 | **6.08** | 11.34    | 9.70 | 12.88 | 11.45 | 8.97 | 6.91     | 9.69     | 6.16 |
> | SVHN      | 2.10     | 1.83     | 1.30 | 2.82  | 2.74  | 1.51 | 1.66     | **1.28** | 1.67 |
> | MNIST     | 0.21     | **0.14** | 0.16 | 0.37  | 0.31  | 0.24 | **0.14** | 0.34     | 0.30 |
> | FMNIST    | 4.00     | 3.61     | 4.12 | 4.53  | 4.65  | 3.39 | **3.26** | 4.50     | 3.32 |
> | ImageNet  | **2.05** | 3.07     | 2.64 | -     | 4.75  | 4.26 | 2.71     | -        | -    |
>
> ---
> **(2) Why OT-induced regions often give high-confidence mispredictions?**
> Intuitively, our semi-discrete OT map induces Laguerre cells whose shared faces are transition regions where the optimal transport switches from one class prototype to another, so multiple classes compete there. These regions contain very few training samples, so the loss almost never directly penalizes mistakes there. With a standard smooth network and loss, the classifier tends to extrapolate the high-confidence predictions it has learned for nearby, well-sampled points into these unlabeled transition regions, even though the correct class should change. As a result, points near OT-induced interfaces are natural locations for high-confidence mispredictions.
>
> As quantitative evidence, the table below reports Fréchet inception distance (FID) to CIFAR-10 (ID) and the mean maximum confidence (MMC) of a ResNet-18 classifier trained on CIFAR-10, evaluated on the boundary samples (OOD) used by CEDA, ACET, CODES, VOS, OE, and OTIS, with results on the CIFAR-10 test set included for reference.
>
> | Dataset  | FID  | MMC   |
> |---------|------|-------|
> | CIFAR-10| 0.00 | 97.98 |
> | CEDA    | 7.25 | 87.39 |
> | ACET    | 7.49 | **99.98** |
> | CODES   | 3.18 | 88.84 |
> | VOS     | 5.28 | 88.51 |
> | OE      | 5.73 | 84.70 |
> | OTIS    | **2.45** | 91.29 |
>
> OTIS attains the smallest FID to CIFAR-10 (2.45 vs. 3.18–7.49 for other methods), meaning its samples stay closest to the ID distribution, while still maintaining high MMC (91.29), clearly higher than CEDA, CODES, VOS, and OE. Although ACET achieves the highest MMC (99.98), the Adv. Noise it uses lies much farther from CIFAR-10 (FID = 7.49). Thus OTIS focuses on near-distribution, high-confidence regions that prior schemes under-cover, which aligns with the above intuition and explains why OT-induced transition regions are particularly prone to high-confidence mispredictions.

---

### Official Review · Reviewer_u5rM · 2025-11-01

**Soundness:** 2
**Presentation:** 3
**Contribution:** 2
**Rating:** 4
**Confidence:** 3

**Summary:**

This work proposes a method to reduce a classifier's confidence on out-of-distribution (OOD) samples. The high-level idea is to construct a set of samples on the semantic boundary, and then train the model to have low confidence on this set of samples. The authors propose to use semi-discrete optimal transport (OT) to construct the boundary. Specifically, the authors sample a set of ID samples, use an encoder to get their latent features, and get the boundary for each pair of samples using Eqn. (9). The authors test their method on some vision datasets.

**Strengths:**

1. The writing is fine. It is not hard to understand the paper
2. The description of the OT method is clear

**Weaknesses:**

1. Motivation is not very clear. It is a natural idea to lower the model's confidence on boundary samples in order to lower its confidence on OOD samples. However, it is not clear to me why using the specific method proposed in this work. There seems to be a bunch of possible ways to construct the boundary samples, and the authors do not elaborate on why the specific method proposed here is more efficient. While the formulation looks reasonable, it looks quite complicated and I am not really sure what this extra complication buys us.
2. Some important details are unclear. I might have missed it, but the two important details I fail to find in the paper are (a) how the encoder in Eqn. (6) should be trained, and (b) how the $n$ training samples after Eqn. (7) should be drawn. I think these two details are particularly important to the success of the propose method, so some extra elaboration is needed.
3. Regarding the experiments, from Table 1 I notice that the proposed method always has a high test error, so it is not clear to me if the decrease in the confidence on OOD samples actually comes from this trade-off. I also think that a brief description of the compared baselines is necessary.

**Questions:**

1. How is the encoder in Eqn. (6) trained?
2. If the encoder is not well trained, how will it affect the performance of your method?
3. How to choose $x_1,\cdots,x_n$ in line 194?

---

> ### Author Response · Authors · 2025-11-21
> **Author Rebuttal Part (1/2)**
>
> We sincerely appreciate the reviewer’s thoughtful feedback and now respond to the main points in detail.
>
> **(1) Necessity of the OT framework beyond heuristic boundary sampling.**
> We agree that lowering confidence on boundary samples is natural; our method aims to *systematically* find the most harmful ones: samples that stay close to the ID feature distribution yet still receive very high confidence. Semi-discrete OT provides a global partition whose interfaces mark where the transport assignment switches between class prototypes, so sampling along these interfaces directly targets low-density, ambiguous regions near the ID manifold, rather than relying on local or ad-hoc boundary heuristics.
>
> The table below reports Fréchet inception distance (FID) to CIFAR-10 (ID) and the mean maximum confidence (MMC) of a ResNet-18 classifier trained on CIFAR-10, evaluated on the boundary samples (OOD) used by CEDA, ACET, CODES, VOS, OE, and OTIS, with results on the CIFAR-10 test set included for reference.
>
> |Dataset|FID|MMC|
> |-|-|-|
> |CIFAR-10|0.00|97.98|
> |CEDA|7.25|87.39|
> |ACET|7.49|**99.98**|
> |CODES|3.18|88.84|
> |VOS|5.28|88.51|
> |OE|5.73|84.70|
> |OTIS|**2.45**|91.29|
>
> OTIS attains the smallest FID to CIFAR-10 (2.45 vs. 3.18–7.49 for other methods), meaning its samples are closest to the ID distribution, while still maintaining high MMC (91.29), clearly higher than CEDA, CODES, VOS, and OE. Although ACET achieves the highest MMC (99.98), its adopted Adv. Noise lies much farther from CIFAR-10 (FID = 7.49). Thus OTIS focuses on near-distribution, high-confidence regions that prior schemes under-cover, providing more informative OOD training signals and explaining the stronger OOD MMC improvements observed in our experiments.
>
> ---
> **(2) Clarity of implementation details.**
> As described in Sec. 4.1 (Implementation), the encoder in Eq. (6) and the decoder in Eq. (7) are jointly trained as a standard autoencoder on the entire ID training set. For small images (28×28 or 32×32) we use a five-layer convolutional encoder with 512 channels and a symmetric five-layer transposed-convolution decoder, yielding a 256-dimensional latent space; for ImageNet (224×224) we use a symmetric VGG-16-based autoencoder with a 1024-dimensional latent space. All autoencoders are trained for 200 epochs with Adam (lr = 1e-4) and then frozen when constructing OTIS samples and training the classifier.
>
> In Eq. (7), $x_1,\dots,x_n$ denote all $n$ images in the ID training set (no subsampling); their latent codes serve as the full support of the discrete OT target measure in the semi-discrete OT step.
>
> ---
> **(3) Whether improved OOD confidence comes from sacrificing ID performance?**
> We agree that OTIS shows a mild trade-off: on some datasets the test error is slightly higher than the base model, but the gap is small and comparable to other OOD overconfidence mitigation methods, while the gains in OOD robustness (OOD MMC, FPR95, AUROC) are much larger.
> To further substantiate that OTIS alleviates OOD overconfidence without sacrificing ID performance, we report ID expected calibration error (ECE, ×10^{-2}, lower is better) in the table below. OTIS achieves the best ECE on CIFAR-10 and FMNIST and remains close to the baseline on ImageNet, indicating that it reduces OOD overconfidence while maintaining well-calibrated predictions on ID data.
>
> |Dataset|Base|CEDA|ACET|CCUs|CODEs|VOS|Ours|OE|CCUd|
> |-|-|-|-|-|-|-|-|-|-|
> |CIFAR-10|3.77|4.61|3.73|3.81| 4.26|2.95|**1.88**|3.65|2.91|
> |CIFAR-100|**6.08**|11.34|9.70|12.88|11.45|8.97|6.91|9.69|6.16|
> |SVHN|2.10|1.83|1.30|2.82|2.74|1.51|1.66|**1.28**|1.67|
> |MNIST|0.21|**0.14**|0.16|0.37|0.31|0.24|**0.14**|0.34|0.30|
> |FMNIST|4.00|3.61|4.12|4.53|4.65|3.39|**3.26**|4.50|3.32|
> |ImageNet|**2.05**|3.07|2.64|-|4.75|4.26|2.71|-|-|

---

> ### Author Response · Authors · 2025-11-21
> **Author Rebuttal Part (2/2)**
>
> **(4) Effect of an imperfect encoder–decoder.**
> We investigate this by varying only the depth of the symmetric encoder–decoder within the same OTIS pipeline and objectives: a shallow 3-layer architecture, our default 5-layer one, and a deeper 7-layer one (each with its own autoencoder and OT partition). As shown in the tables below, all three variants achieve similar test error and ID ECE, and their OOD metrics (MMC, FPR95, AUROC) remain close; the 5-layer architecture provides the best overall balance, while the deeper 7-layer model does not consistently help and sometimes slightly worsens OOD scores, likely due to mild overfitting. This suggests that OTIS is robust to reasonable variations in autoencoder capacity and does not require an extremely deep encoder–decoder; only if the autoencoder were very poorly trained and failed to produce a class-structured latent space would the benefit of OTIS naturally diminish.
>
> Table 1. TE, ECE, and MMC on the CIFAR-10 ID test set.
> | Method  | TE | ECE | MMC |
> |-|-|-|-|
> | Shallow | 7.60        | 2.45         | **96.00**    |
> | Deep    | 8.55        | 2.55         | 92.31        |
> | Default (Ours)| **7.52**    | **1.88**     | 95.46        |
>
> Table 2. MMC (%, ↓) on various OOD datasets.
> | Method  | SVHN | CIFAR-100 | LSUN_CR | Textures_C | Noise | Uniform | Adv.Noise | Adv.Samples | CIFAR-10-C |
> |-|-|-|-|-|-|-|-|-|-|
> | Shallow | 24.95 | 67.74 | 34.47 | 56.88 | 28.70 | **10.00** | 44.51 | 60.21 | 73.85 |
> | Deep    | 14.27 | **59.83** | 34.72 | 54.57 | 21.59 | 10.04    | 31.37 | 59.63 | **64.97** |
> | Default (Ours)| **13.18** | 64.79 | **30.36** | **48.75** | **16.18** | **10.00** | **26.42** | **57.71** | 70.78 |
>
> Table 3. FPR95 (%, ↓) on various OOD datasets.
>
> | Method  | SVHN | CIFAR-100 | LSUN_CR | Textures_C | Noise | Uniform | Adv.Noise | Adv.Samples | CIFAR-10-C |
> |-|-|-|-|-|-|-|-|-|-|
> | Shallow | 3.11 | 65.34 | 19.27 | 46.22 | 0.23 | **0.00** | 3.00  | 77.55 | 68.60 |
> | Deep    | 1.23 | 68.63 | 22.77 | 50.07 | 4.54 | **0.00** | 1.09  | 78.64 | 66.96 |
> | Default (Ours)| **1.21** | **63.38** | **10.95** | **36.70** | **0.00** | **0.00** | **0.66** | **74.71** | **66.16** |
>
> Table 4. AUROC (%, ↑) on various OOD datasets.
> | Method  | SVHN | CIFAR-100 | LSUN_CR | Textures_C | Noise | Uniform | Adv.Noise | Adv.Samples | CIFAR-10-C |
> |-|-|-|-|-|-|-|-|-|-|
> | Shallow | 99.27    | 86.78    | 97.07    | 91.42    | 99.48    | **100.00** | 91.96    | 60.97    | 75.87    |
> | Deep    | 99.50    | 84.91    | 96.02    | 91.38    | 97.26    | **100.00** | 92.07    | 64.47    | 75.46    |
> | Default (Ours) | **99.79**| **87.19**| **98.26**| **93.01**| **99.97**| **100.00** | **95.78**| **67.27**| **76.63**|

---

> ### Author Response · Authors · 2025-11-27
> **Follow-up on our rebuttal and revision**
>
> Dear Reviewer #u5rM,
>
> Thank you very much for your thoughtful and constructive comments on our paper. We have carefully addressed your comments in our rebuttal and revised the main paper accordingly. We would greatly appreciate it if you could take a look and let us know whether there are any remaining concerns. We look forward to further discussion and are happy to clarify any remaining questions during the discussion phase.
>
> Thank you for your time and kind consideration.
>
> Best regards,
>
> The authors

---

### Official Review · Reviewer_rQDu · 2025-11-01

**Soundness:** 3
**Presentation:** 2
**Contribution:** 3
**Rating:** 4
**Confidence:** 4

**Summary:**

This paper addresses the general issue of deep neural networks producing overconfident predictions on out-of-distribution (OOD) inputs. The authors propose to generate specific proxy OOD samples. The core idea is to identify regions of semantic
ambiguity in the latent space of an AE by leveraging the geometry of semi-discrete Optimal Transport (OT). They solve an OT problem between a base distribution and the latent embeddings of training data, in order to identify singular boundaries. Then, they generate OTIS by interpolating latent features near these boundaries of interest. The paper claims this geometrically grounded approach significantly reduces OOD overconfidence compared to SOTA methods.

**Strengths:**

- the conceptual link drawn between singularities in semidiscrete OT and semantic ambiguity is interesting. Targeting these specific regions for confidence suppression is potentially more founded than using heuristics like noise or generic outlier datasets

- the method for generating OTIS is detailed and the sample creation between latent concepts should provide a more effective signal for training than random interpolations or generic augmentations (e.g. Figure 5)

- strong performance in experiments

**Weaknesses:**

- The paper heavily relies on the theory of semi-discrete OT (Brenier potential, Laguerre cells) in order to define a notion of neighborhood (i.e., adjacency cell boundaries). However, the core measure of ambiguity is simply the angle between neighbor latent vectors, which is independent of the OT framework. Hence, the paper doesn’t clearly explain/justify  the additional complexity : why the OT-neighborhood concept is necessary over a much simpler and standard k-NN approach.

- The primary goal is to mitigate overconfidence, (i.e, lead the model toward better calibrated predictions). While the work shows ID MMC remains high, this doesn’t eliminate  the possibility that the method makes the model uncalibrated on correct ID predictions due to the nature of the loss and the generated OoD inputs. It would be crucial in my opinion to report the ECE score on the ID samples as it may reduce overconfidence on ID inputs, which would open the discussion a little bit.

- I might have missed some things, but there are some major complexity drawbacks potentially. Solving a semi-discrete OT problem for the entire training set (n=50k for CIFAR, n=1.2M for ImageNet) is certainly computationally demanding. Do the authors rely on solving OT on mini-batches of target points {$y_i$} ? If so, the geometric partition (Laguerre cells, boundaries) may be unstable and change with each batch. In this case, the paper should provide a theoretical error bound for this approximated domain/boundary relative to the true partition (over all the dataset for instance), using for instance Hausdorff distance. This may depend on the number of points, the dimension of the latent space and also probably involve some topological assumption [1] on the representativeness of the mini-batch. If not, the paper must provide a clear analysis of the computational complexity and demonstrate the feasibility and runtime of their OT algorithm on large-scale datasets. Not discussing this aspect is detrimental for the readers.

- The method trains the model to reduce confidence on inputs deviating from the clean ID manifold, using semantic ambiguity. This might teach the model to also reduce confidence on inputs suffering from diverse domain shift such as common corruptions (e.g., CIFAR-10-C dataset). As modern architectures are well known to be overconfident on such inputs, it would be interesting to see MMC and ECE score on corruption benchmarks.

[1] Jean-Daniel Boissonnat, Frédéric Chazal, and Mariette Yvinec. Geometric and topological inference. Vol. 57. Cambridge University Press, 2018.

**Questions:**

The work introduces an interesting idea for generating targeted samples to reduce OOD confidence and shows promising results on the OOD MMC metric, however it has some important weaknesses
in my opinion. The theoretical justification based on OT appears too complex for the actual mechanism employed and lacks comparison to natural alternatives (k-NN) and the absence of standard calibration metrics (ECE). The paper’s heavy reliance on dense OT and writing makes it difficult to understand the core mechanism, which is essentially an advanced latent-space mixup. Concerns about scalability/stability and potential negative impacts on robustness further weaken the submission. I am open to reconsidering my score if the authors can provide a convincing rebuttal that addresses these weaknesses, namely :

- clarify the necessity of the OT framework
- clarify the impact on ECE
- clarify the complexity implications
- clarify the standing on corruption/domain shift

**Details Of Ethics Concerns:**

nothing in particular

---

> ### Author Response · Authors · 2025-11-21
> **Author Rebuttal Part (1/2)**
>
> We thank the reviewer for the insightful comments and the willingness to reconsider the score upon clarification. Below, we address the main concerns in detail.
>
> **(1) OT-induced interfaces vs. k-NN neighborhoods.**
> Our goal is not just to find “similar neighbors” but to locate interfaces in latent space that approximate semantic decision boundaries, i.e., regions where the representation should switch from one class prototype to another. Semi-discrete OT provides such a global partition: the Brenier potential induces Laguerre cells, and their shared faces are exactly where the optimal transport assignment switches from prototype $i$ to $j$, which OT theory identifies as structurally unstable and thus naturally ambiguous regions. The angle is then used only to rank *these* OT-induced interfaces by how strongly their transport directions disagree.
>
> In contrast, a k-NN graph is defined purely by local distances: it does not produce separating surfaces, nor does it guarantee that its edges lie near any decision boundary or transport switch, so it cannot play the same role as the OT partition. Empirically, when we replace OT-induced interfaces with random pairs or naive latent mixup (which is closer in spirit to k-NN–style neighborhoods), OOD MMC clearly degrades, indicating that the OT geometry is essential to concentrating samples near truly ambiguous regions rather than being a cosmetic layer on top of a cosine-based score.
>
> ---
> **(2) Necessity of the OT framework beyond heuristic boundary sampling.**
> Our motivation is to explicitly target the most harmful errors: samples that stay very close to the ID distribution while still receiving high confidence from the classifier. These “near-distribution, high-confidence” points are where OOD overconfidence is most critical yet hardest to expose. While ACET (via its adopted Adv. Noise), CEDA, CODES, VOS, and OE all build auxiliary boundary sets, semi-discrete OT provides a principled global geometry: the Brenier potential partitions the latent space, and the interfaces where the transport assignment switches between class prototypes mark low-density transition zones around the ID manifold. Sampling along these interfaces allows OTIS to concentrate on such ambiguous regions without relying on ad-hoc local heuristics.
>
> The table below reports Fréchet inception distance (FID) to CIFAR-10 (ID) and the mean maximum confidence (MMC) of a ResNet-18 classifier trained on CIFAR-10, evaluated on the boundary samples (OOD) used by CEDA, ACET, CODES, VOS, OE, and OTIS, with results on the CIFAR-10 test set included for reference.
>
> |Dataset|FID|MMC|
> |-|-|-|
> |CIFAR-10|0.00|97.98|
> |CEDA|7.25|87.39|
> |ACET|7.49|**99.98**|
> |CODES|3.18|88.84|
> |VOS|5.28|88.51|
> |OE|5.73|84.70|
> |OTIS|**2.45**|91.29|
>
> OTIS attains the smallest FID to CIFAR-10 (2.45 vs. 3.18–7.49 for other methods), meaning its samples are closest to the ID distribution, while still maintaining high MMC (91.29), clearly higher than CEDA, CODES, VOS, and OE. Although ACET achieves the highest MMC (99.98), its adopted Adv. Noise lies much farther from CIFAR-10 (FID = 7.49). Thus OTIS focuses on near-distribution, high-confidence regions that prior schemes under-cover, providing more informative OOD training signals and explaining the stronger OOD MMC improvements observed in our experiments.
>
>
> ---
> **(3) Calibration of in-distribution predictions (ECE).**
> Following the suggestion, we report ID expected calibration error (ECE) values (×1e-2, lower is better) in the table below. OTIS achieves the best ECE on CIFAR-10 and FMNIST and remains close to the baseline on ImageNet, indicating that it reduces OOD overconfidence while maintaining well-calibrated predictions on ID data.
>
> |Dataset|Base|CEDA|ACET|CCUs|CODEs|VOS|Ours|OE|CCUd|
> |-|-|-|-|-|-|-|-|-|-|
> |CIFAR-10|3.77|4.61|3.73|3.81| 4.26|2.95|**1.88**|3.65|2.91|
> |CIFAR-100|**6.08**|11.34|9.70|12.88|11.45|8.97|6.91|9.69|6.16|
> |SVHN|2.10|1.83|1.30|2.82|2.74|1.51|1.66|**1.28**|1.67|
> |MNIST|0.21|**0.14**|0.16|0.37|0.31|0.24|**0.14**|0.34|0.30|
> |FMNIST|4.00|3.61|4.12|4.53|4.65|3.39|**3.26**|4.50|3.32|
> |ImageNet|**2.05**|3.07|2.64|-|4.75|4.26|2.71|-|-|

---

> ### Author Response · Authors · 2025-11-27
> **Author Rebuttal Part (2/2)**
>
> **(4) Complexity and scalability.**
> In all our experiments we solve a single *global* semi-discrete OT problem on the full training set in latent space, rather than approximating it on mini-batches. Because OT is computed on compressed latent features instead of raw pixels, the latent dimensionality is moderate and the global OT step fits comfortably in GPU memory (≈0.3 GB for CIFAR-10 and ≈12.5 GB for ImageNet on a 24 GB GPU).
>
> We appreciate this insightful comment. On larger datasets GPU memory can indeed become a bottleneck, and exploring more compact latent embeddings or mini-batch / distributed OT solvers that still approximate the global OT partition well is an important direction for future work.
>
> ---
> **(5) Performance on corruption benchmarks.**
> Following the suggestion, we evaluate OTIS on CIFAR-10-C and CIFAR-100-C; the table below shows that OTIS substantially lowers MMC and FPR95 while improving AUROC compared to the base model and most baselines. This indicates that training on semantically ambiguous OTIS samples also mitigates overconfidence and improves detection under common corruptions.
>
> |Method|CIFAR-10-C MMC|CIFAR-10-C FPR95|CIFAR-10-C AUROC|CIFAR-100-C MMC|CIFAR-100-C FPR95|CIFAR-100-C AUROC|
> |-|-|-|-|-|-|-|
> |Base|93.21|83.30|70.43|65.61|84.06|68.92|
> |CEDA|86.81|80.92|70.66|69.85|89.30|63.17|
> |ACET|90.81|85.00|66.16|71.80|87.40|65.62|
> |CCUs|85.42|80.96|66.16|66.19|79.78|65.94|
> |CODES|84.44|84.80|68.00|74.96|85.80|67.04|
> |VOS|90.89|82.38|66.07|70.90|84.72|67.67|
> |Ours|**70.78**|**66.16**|**76.63**|**59.22**|**73.82**|72.72|
> |OE|87.75|79.86|70.15|71.18|85.20|66.16|
> |CCUd|83.82|68.79|73.99|63.27|78.16|**79.20**|

---

> ### Author Response · Authors · 2025-11-27
> **Follow-up on our rebuttal and revision**
>
> Dear Reviewer #rQDu,
>
> Thank you very much for your thoughtful and constructive comments on our paper. We have carefully addressed your comments in our rebuttal and revised the main paper accordingly. We would greatly appreciate it if you could take a look and let us know whether there are any remaining concerns. We look forward to further discussion and are happy to clarify any remaining questions during the discussion phase.
>
> Thank you for your time and kind consideration.
>
> Best regards,
>
> The authors

---

### Author Response · Authors · 2025-11-21
**Summary of Our Response and Main Updates**

Dear Area Chair and Reviewers,

We thank the Area Chair and the Reviewers for their insightful feedback. We are encouraged that one reviewer provided a very positive assessment and that another stated they would be open to reconsidering their score if the weaknesses were convincingly addressed. The remaining reviewers mainly ask for clarification and implementation details, which we address in our responses and reflect in the revised manuscript. In view of the updated ICLR review process, we focus on providing a clear record of these clarifications and additional results to assist your assessment.

**High-level summary for the Area Chair.**
Our work addresses out-of-distribution (OOD) overconfidence. Instead of heuristic proxy OOD constructions, we exploit the geometry of semi-discrete optimal transport (OT): transport singularities identify structurally unstable regions where the Brenier potential is non-differentiable and class assignments are ambiguous. We construct OT-induced OOD samples (OTIS) near these singularities and apply confidence suppression during training, yielding a principled way to regularize confidence in these ambiguous low-density regions, rather than uniformly lowering confidence.

Below we summarize the main updates to the paper and how they address the core concerns:

1. **Necessity of the OT framework beyond heuristic boundary sampling (Sec. 4.3).**
The original experiments have showed that OTIS consistently improves OOD calibration over CEDA, ACET, CODES, VOS, and OE. Reviewers, however, questioned whether these gains truly require the semi-discrete OT framework, or whether similar performance could be obtained with simpler boundary constructions. To clarify what OT contributes beyond simple boundary heuristics, we add a quantitative comparison of the boundary samples used by CEDA, ACET, CODES, VOS, OE, and OTIS, reporting Fréchet inception distance (FID) to in-distribution (ID) data and mean maximum confidence (MMC) of the base classifier on these samples.
    The results show that OTIS concentrates on near-distribution, high-confidence regions that heuristic schemes largely miss: the other methods’ samples either drift far from the ID manifold or mostly have low confidence. This indicates that the stronger OOD calibration comes from sampling OT-induced transition regions, rather than from generic boundary sampling or additional tuning.


2. **Clarifying the trade-off with ID calibration (Sec. 4.3).**
Reviewers worried that the reduction in OOD confidence might simply come from degrading the classifier or uniformly lowering confidence on all inputs. We therefore report expected calibration error (ECE) on ID test sets in addition to OOD metrics. The added results show that OTIS alleviates OOD overconfidence while preserving, and in some cases improving, calibration on ID data, indicating that the gains are not obtained by globally weakening the classifier but by targeting overconfidence in ambiguous regions.

3. **Robustness to the choice of base distribution (Sec. 4.3).**
Reviewers also asked whether our conclusions depend on using a particular base distribution in the latent space. We thus compare Gaussian and uniform bases under the same autoencoder architecture. OTIS is robust to this choice, and the uniform base serves as a strong default, alleviating concerns about sensitivity to this design decision.

4. **Robustness to autoencoder depth (Sec. 4.3).**
It was further unclear whether our findings might be tied to a specific encoder–decoder architecture. We therefore evaluate shallow, medium, and deep autoencoders with a uniform base. OTIS remains effective across depths, and the 5-layer symmetric encoder–decoder offers a good balance between computational cost and the effectiveness in alleviating OOD overconfidence, indicating that our conclusions do not rely on a carefully tuned architecture.

5. **Performance under common corruptions (Appendix A.1).**
Reviewers also asked whether the improvements would hold under more realistic distribution shifts beyond synthetic or specially constructed OOD sets. We therefore assess OTIS on corruption-based OOD scenarios by treating CIFAR-10/100 as ID data and CIFAR-10-C/100-C as OOD inputs. OTIS significantly reduces OOD MMC and FPR95 and improves AUROC over baselines, showing that the benefits extend to practical corruption-based shifts.

6. **Correction of the interpolation weight $\lambda_i$ (Sec. 4.1).**
The previous description of the interpolation weight $\lambda_i$ could be read as inconsistent with its actual definition in the method. We corrected the text so that it is fully consistent with its definition and usage in the main part of the paper, improving clarity and avoiding potential confusion.

We sincerely appreciate the time and effort that the Area Chair and the Reviewers devoted to evaluating our work and helping us improve the paper. Thank you very much for your consideration.

Best regards,
The authors

---

### Meta-Review · Area_Chair_khAm · 2025-12-19

**Summary:**

Reviewers raised three main concerns:

1/ [/] Whether the theoretical justification based on optimal transport was necessary to explain the actual algorithm (rQDu, u5rM). Authors agreed that the paper did not have formal mathematical guarantees, but showed empirically that it generated samples that were closer to the training and where the model was overly confident.

2/ [+] Whether the method degraded in-distribution calibration (rQDu, u5rM). Authors addressed this point by providing measures of in-distribution calibration (ECE).

3/ [-] High test error on the in-distribution test set (u5rm, QHM5) It seems like the author's main rebuttal was that the method is _calibrated_ on in-distribution. However, I'm not sure that this addresses the concern that the test accuracy is poor, not only compared to the base model but also compared to some of the baselines (esp. VOS). E.g., CIFAR100: 23% -> 28% error; SVHN: 3.7% -> 4.8%; FMNIST: 6.9% -> 7.4%.

Overall, the paper presents a new perspective (via OT) for tackling overconfidence on OOD samples. I think this idea may be of interest to the broader ICLR community, and hence recommend that the paper be accepted.

I would encourage the authors to emphasize the somewhat poor in-distribution test error as a limitation of the paper. To help underscore this, I'd recommend that Table 1 be revised so that the Test Error rows (labeled "TE") have the best entries bolded as well.

**Reviewer Concerns:**

(see below)

**Reviewer Scores:**

rQDu: 4 --> 6
* [+] why the OT-neighborhood concept is necessary over a much simpler and standard k-NN approach: authors seem to have addressed with comparisons to random pairs and naive latent mixup
* [+] report the ECE score on the ID samples and corruption benchmarks: authors addressed with additional experiments
* [/] computational cost: authors reported memory usage; it would have been nice to see time measured as well.

u5rM: 4 --> 4
* [/] "There seems to be a bunch of possible ways to construct the boundary samples ... I am not really sure what this extra complication buys us.": Authors show that their method generates samples that are more ambiguously labeled and more in-distribution, as compared with alternatives. The authors don't provide an argument for why there might not exist a simpler method, but I'm not sure that's a reasonable question to expect the authors to answer.
* [+] how the encoder in Eqn. (6) should be trained, and (b) how the  training samples after Eqn. (7): authors provided details during the rebuttal.
* [-] high test error: not really addressed
* [-] description of baselines: not addressed.

QHM5: 8 --> 8
* [/] high test error: authors argued that model is calibrated in-distribution.

mC83: 4 --> 6
* [/] The result is a strong intuition with empirical support rather than a theorem: authors acknowledge this concern
* [+] hard coded weight $\lambda_i$: authors fixed this typo
* [-] missing experimental details: not addressed
* [+] Does the method suffer from overconfidence on in-distribution samples?

---

### Decision · Program_Chairs · 2026-01-26

Accept (Poster)